# *Harbinger* transposon insertion in ethylene signaling gene leads to emergence of new sexual forms in cucurbits

Hsin-Ya Huang [1,3], Siqi Zhang [1,3], Fadi Abou Choucha [1,3], Marion Verdenaud[1], Feng-Quan Tan[1], Clement Pichot[1], Hadi Shirazi Parsa[1], Filip Slavkovic [1], Qinghe Chen[1], Christelle Troadec [1], Fabien Marcel[1], Catherine Dogimont [2], Leandro Quadrana[1], Adnane Boualem [1] & Abdelhafid Bendahmane [1] ✉

In flowering plants, the predominant sexual morph is hermaphroditism, and the emergence of unisexuality is poorly understood. Using *Cucumis melo* (melon) as a model system, we explore the mechanisms driving sexual forms. We identify a spontaneous mutant exhibiting a transition from bisexual to unisexual male flower, and identify the causal mutation as a *Harbinger* transposon impairing the expression of *Ethylene Insensitive 2* (*CmEIN2*) gene. Genetics and transcriptomic analysis reveal a dual role of *CmEIN2* in both sex determination and fruit shape formation. Upon expression of *CmACS11*, EIN2 is recruited to repress the expression of the carpel inhibitor, *CmWIP1*. Subsequently, EIN2 is recruited to mediate stamina inhibition. Following the sex determination phase, EIN2 promotes fruit shape elongation. Genome-wide analysis reveals that *Harbinger* transposon mobilization is triggered by environmental cues, and integrates preferentially in active chromatin, particularly within promoter regions. Characterization of a large collection of melon germplasm points to active transpositions in the wild, compared to cultivated accessions. Our study underscores the association between chromatin dynamics and the temporal aspects of mobile genetic element insertions, providing valuable insights into plant adaptation and crop genome evolution.

Flowering plants exhibit a remarkable diversity of reproductive strategies, ranging from asexual to crossbreeding, each strategy influencing the genetic landscape and evolutionary trajectory of plant populations. One crucial aspect of this reproductive diversity lies in the development of unisexual flowers, a phenomenon intricately linked to sex determination mechanisms[1]. In dioecious species, plants bear either male or female flowers. In monoecious species, separate male and female flowers develop on the same individual. Other intermediary systems also exist. Andromonoecious plants bear both male and bisexual flowers on the same plants, and

gynoecious and androecious plants bear only female and male flowers, respectively[1,2].

Two models have been proposed to explain the transition to unisexuality. The first model, known as sex allocation theory, posits that ecological and physiological factors favor directing resources toward a single sexual function rather than both[3,4]. The second model relies on population genetic arguments, suggesting that dioecy becomes advantageous in the presence of inbreeding depression[5,6]. Consistent with these concepts, in monoecious species, the female function is prioritized when resources are ample, while the male

[1]Université Paris-Saclay, CNRS, INRAE, Université Evry, Institute of Plant Sciences Paris-Saclay (IPS2), 91190 Gif-sur-Yvette, France. [2]INRAE, Génétique et Amélioration des Fruits et Légumes (GAFL), 84143 Montfavet, France. [3]These authors contributed equally: Hsin-Ya Huang, Siqi Zhang, Fadi Abou Choucha. ✉e-mail: Abdelhafid.bendahmane@inrae.fr

function is expressed more often under adverse conditions, or when the female function is already fulfilled[7]. Male plants excel in pollen donation, potentially attributed to their capacity to produce a greater quantity of male flowers and pollen[8]. Besides, pollen development is highly sensitive to adverse temperatures[9]. Producing more male flowers could serve as a compensatory mechanism, mitigating the impact of the adverse conditions on male function.

In spite of our progress in decoding the genetic pathway leading to the development of unisexual flowers[10–18], the molecular mechanisms integrating environmental cues driving the emergence of new sexual morphs are still obscure. The tapestry of evolutionary processes is woven with the intricate threads of genetic innovation. Among the most enigmatic contributors are transposable elements (TEs), ubiquitous mobile genetic elements that traverse the genomes of organisms across the tree of life[19]. Based on their mobility, TEs can be classified as DNA transposons, which use a cut-and-paste mechanism for their mobilization, and retro-transposons, which jump via a reverse-transcribed RNA intermediate. These two classes are further divided into TE superfamilies and families based on particular sequences, such as the presence of specific terminal repeats or conserved protein domains[19]. For instance, *Harbinger* transposons are DNA transposons which usually encode a DDE transposase and a SANT/Myb/trihelix domain-containing DNA-binding protein[20]. The role of transposons in adaptation and genome evolution is complex and multifaceted. Transposons can cause genetic instability by disrupting the functions of essential genes. Transposons can also contribute to genome evolution by creating advantageous genetic modifications[21].

In the present work, we investigated how sexual morphs emerge in *Cucurbitaceae*, a large plant family that displays different sexual morphs. We focused our investigation on a monoecious species, *Cucumis melo* (melon). Male flowers are prevalent in melon, while female flowers develop on the initial nodes of the branches. During early developmental stages, flowers are initially bisexual. Subsequently, sex determination occurs through the developmental arrest of either stamen or carpel primordia[2,18]. This process is controlled by the monoecy pathway implicating the *andromonoecious* (*M*), *androecious* (*A*), and *gynoecious* (*G*) genes. The *G* gene codes for the transcription factor *CmWIP1*, a C2H2 zinc finger transcription factor, which functions as a master regulator in the process of sex determination in cucurbits[16]. The *A* and *M* genes encode for two aminocyclopropane-1-carboxylic acid synthase enzymes, CmACS7 and CmACS11, respectively[14,15]. *CmACS11* expression in lateral branches suppresses the expression of the male-promoting gene *CmWIP1*. *CmACS7* expression in carpel primordia of female flowers inhibits stamina development through a non-autonomous mechanism[17].

Through characterization of a spontaneous sex transition mutant, we identified a *Harbinger* transposon, *AndroPIF*, inserted in the *Ethylene Insensitive 2* (*EIN2*). We investigated the role of *EIN2* in sex determination and fruit development, and the role of TEs in the emergence of sexual morphs in plants.

## Results

### A *Harbinger* transposon insertion in the EIN2 gene is responsible for androecy in CAM106 line

In andromonoecious melon, male flowers develop in inflorescences at every node of the main stem, and in distal nodes of the lateral branches. Hermaphrodite flowers develop at the first two nodes of lateral branches. During seed propagation of an andromonoecious Charentais line, hereafter named CAM_WT, we identified a spontaneous mutant line that is androecious, CAM106, where hermaphroditic flowers are converted to male unisexual flowers (Fig. 1a–c). Furthermore, male inflorescences develop in clusters of about 15 flowers, compared to 6 in the CAM_WT parental line (Fig. 1d). Androecy in melon can arise via the inactivation of *CmACS11* or the melon *CRABS*

*CLAW* gene (*CmCRC*)[15,18]. So, we sequenced the promoter and the coding regions of the two genes in CAM106 but found no mutations, pointing to a new locus controlling sex transition in cucurbits. To characterize the inheritance of such mutation, we crossed CAM106 to an andromonoecious line, CAM_Sister, and phenotyped 10 F1 hybrids and their F2 progenies. CAM_Sister, is genetically related to CAM_WT but not identical (see material and methods). We found all the F1 hybrids andromonoecious, indicating that the CAM106 mutation is recessive. Further phenotyping of 200 F2 plants showed 3:1 segregation of andromonoecy versus androecy, consistent with a single-locus recessive mutation leading to androecy (Supplementary Fig. S1a, b). We refer hereafter to this novel androecious mutation as *a3*. To identify the causal mutation in the CAM106 line, we sequenced bulked-genomic DNA from andromonoecious, and androecious segregant plants, and determined the delta-SNP index. We found a single genomic region on the right end of chromosome 8 completely associated with androecy (Fig. 1e). Fine mapping in a population of 1500 segregants plants, further delimited *a3* to a single gene, *CmEIN2*, encoding the melon ortholog of the *ethylene-insensitive protein 2*[22] (Fig. 1f). Sequence analysis revealed an insertion of a DNA transposon of the *P Instability Factor (PIF)/Harbinger* superfamily[8,20], henceforth termed *AndroPIF* at the 5′ UTR of *CmEIN2* (Fig. 1f, Supplementary Figs. S1c and S2). Genotyping of plants harboring recombination events in the vicinity of *a3*, using a marker specific to the transposon insertion showed complete association of the transposon insertion in *CmEIN2* and androecy (Supplementary Fig. S1d).

Transposable elements are exposed to epigenetic modifications, with consequences on the expression of nearby genes, especially through the genomic spreading of DNA methylation[23]. We examined the DNA methylation status of CAM_WT parental line and CAM106, using whole genome bisulfite sequencing. As expected, strong DNA methylation of *AndroPIF* was observed. However, weak methylation spreading was detected in *AndroPIF* flanking sequences (Supplementary Fig. S3). To test if *AndroPIF* insertion was interfering with *CmEIN2* transcript accumulation, we measured the expression of *CmEIN2*, using quantitative real-time PCR. We found a weak to no expression of *CmEIN2* in leaves and flowers of CAM106, indicating that *a3* is likely a loss-of-function allele of *CmEIN2* (Fig. 1g).

### Induced mutations in EIN2 lead to androecy in melon

Since there is no efficient transformation protocol in cucurbits, to further validate the role of *CmEIN2* in androecy, we used TILLING (Targeting Induced Local Lesions IN Genomes) collections from monoecious melon and screened for mutations in *CmEIN2*. Sixty-two induced mutations were identified, including two splicing, *ein2-SP1* (G1175A) and *ein2-SP2* (G1246A), and 17 missense mutations. As expected, missense mutations had no effect on the plant sexual phenotype (Supplementary Table 1), consistent with earlier reports on *EIN2* mutagenesis in *Arabidopsis*[22]. We selected the two spicing mutants for further analysis. *ein2-SP1* is altered in the splicing donor site of intron 4, while *ein2-SP2* harbors a mutation in the splicing acceptor site of intron 4 (Supplementary Fig. S4a). Transcript analysis revealed introns 3 and 4 retention (30%), intron 4 retention (47%) or exon 4 skipping (23%) for *ein2-SP1* mutant; similarly, *ein2-SP2* mutation leads to intron 4 retention (27%) or to splice variant with a 4 bp deletion in exon 4 (73%). For both mutants, all the splicing forms lead to premature stop codons and thus are likely loss-of-function alleles (Supplementary Fig. S4b). We backcrossed the mutant lines harboring the splicing mutations to the wild type and followed the segregation of the mutations with the phenotypes. As predicted, plants homozygous for the splicing mutations were androecious, i.e., female flowers being transformed into male flowers (Fig. 2a, b; Supplementary Fig. S5). Interestingly, in plants heterozygous for *ein2-SP1* and *ein2-SP2* mutations, 20-40% of female flowers were transformed into bisexual flowers (Fig. 2c, d;

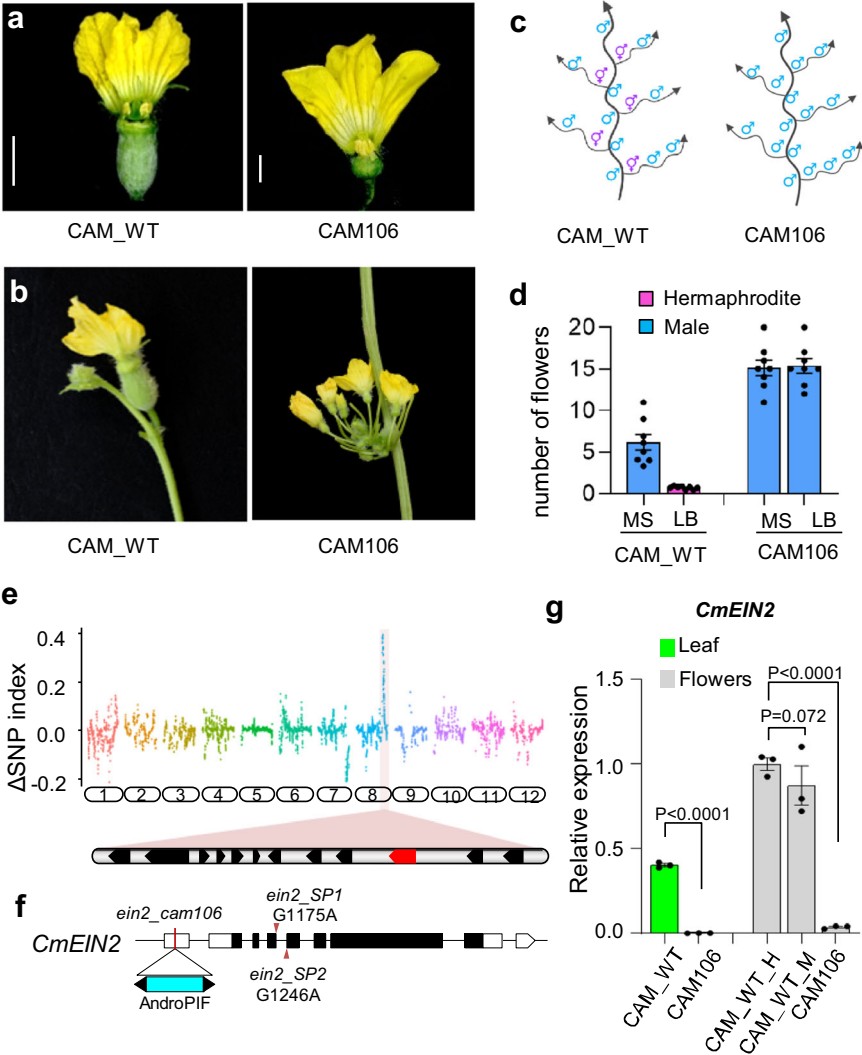

**Fig. 1 | A *Harbinger* transposon insertion in *EIN2* gene is responsible for androecy. a** Sexual phenotype of andromonoecious melon (CAM_WT), and androecious mutant (CAM106). Scale bar = 1 cm. **b** Lateral branches showing a hermaphrodite flower in CAM_WT, and a cluster of male flowers in CAM106. **c** Schematic of sexual morphs. **d** Number of hermaphroditic or male flowers per node on the main stem (MS), or at the 1st node of lateral branches (LB). Data are mean ± SE. (n = 8 biological replicates). **e** Cloning of the *a3* locus. Δ SNP index between andromonoecious and androecious bulks, calculated with a 1-Mb sliding window. Arrows indicate candidate genes on chromosome 8. **f** Structure of the *CmEIN2* gene and the position of *AndroPIF* insertion. The *ein2_SP1* and *ein2_SP2* splicing mutations are shown by arrowhead. **g** Quantitative real-time PCR (qPCR) of *CmEIN2* in leaf, hermaphrodite (H) and male (M) flowers of CAM_WT, and CAM106. *P*-values from an unpaired two-tailed Student's t-test are indicated. Data are mean ± SE. (*n* = 3 biological replicates). Source data are provided as a Source Data file.

Supplementary Fig. S5), highlighting also the role of *CmEIN2* in stamen inhibition in female flowers. This phenotype was not observed in CAM106 mutant because CAM_WT parental line is andromonoecious developing male and hermaphrodite flowers (Fig. 1a).

Since *EIN2* is a key regulator of ethylene signaling[24], we examined the ethylene response phenotype of 3 days old etiolated seedlings of CAM106-AndroPIF insertion mutant and *ein2-SP1* and *ein2-SP2* splicing mutants. Dark-grown melon seedlings treated with ethylene display a typical "triple response" consisting of inhibition of root development and hypocotyl elongation, radial swelling of the hypocotyl, and apical hook formation[25]. We found CAM106, *ein2-SP1* and *ein2-SP2* mutants to be insensitive to ethylene (Fig. 2e), supporting that the *AndroPIF* insertion and the splicing mutations are likely *CmEIN2* loss-of-function alleles. Altogether, the characterization *CmEIN2-AndroPIF* and the *CmEIN2*-splicing mutants validate the role of *CmEIN2* in the development of carpel-bearing flowers, both in andromonoecious and monoecious plants.

## CmEIN2 is required both for carpel development and stamina inhibition

To integrate the function of *CmEIN2* in the monoecy sex determination pathway, we generated mutant combinations with *CmWIP1*, *CmACS7*, and *CmACS11*, and examined the sex morphs of these double mutants in relation with the expression of sex genes (Fig. 3a). In melon, the inappropriate sexual organs are arrested at stage 6 after the initiation of both carpel and stamen primordia[18]. In monoecious plants, expression of *CmACS11* in flower buds of lateral branches represses the expression of the male-promoting gene *CmWIP1*, leading to female flower development. Since loss of function of *CmEIN2* leads to the development of male instead of female flowers in lateral branches, we first analyzed the expression of *CmACS11*, *CmACS7*, and *CmWIP1*. Compared to WT, the expression of *CmACS11* was not compromised. In contrast, the repression of *CmWIP1* expression was relieved, leading to male plants, and indirectly to repression of *CmACS7* that relies on carpel development for its expression (Fig. 3a, b). To investigate the role of *CmEIN2* in stamina inhibition, we crossed *ein2* mutant to *wip1*

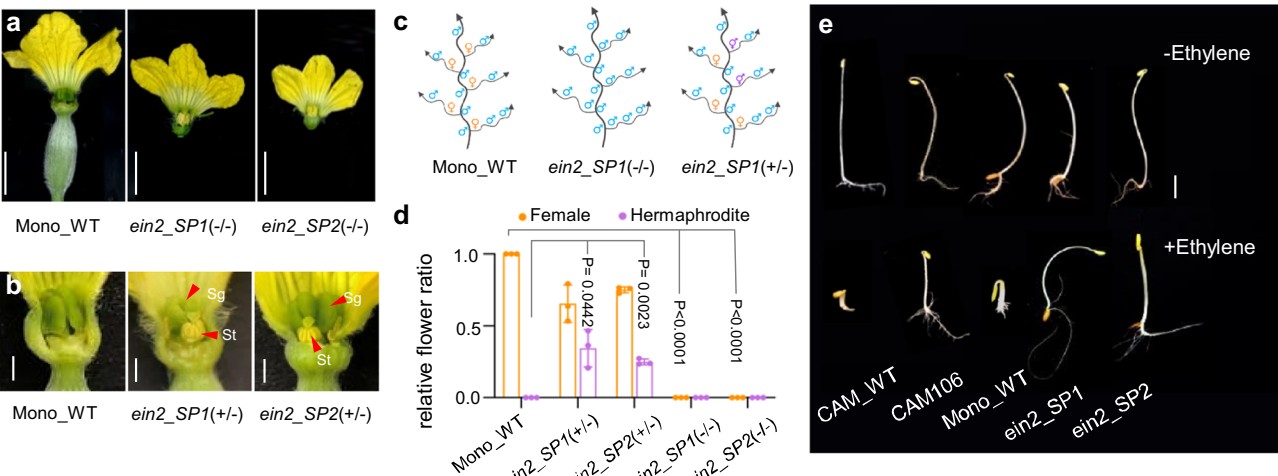

**Fig. 2 | Induced mutations in *CmEIN2* lead to androecy in melon. a** Sexual phenotype of a monoecious plant (Mono_WT), and ein2 splicing mutants (−/−). **b** Sexual phenotype of pistillate flowers of WT and *ein2* heterozygous mutants (+/−). Sg, stigma; st, stamen. Scale bars in (**a**, **b**) = 0.5 cm. **c** Schematic of sexual morphs. **d** Relative ratio of female, or hermaphrodite flowers, per total carpel-bearing flowers. Data are mean ± SE. (*n* = 3 biological replicates). **e** Triple response phenotype of melon genotypes in absent (−Ethylene), and presence of ethylene (+Ethylene). The photography is taken 7 days after germination. Scale bars = 1 cm. Source data are provided as a Source Data file.

gynoecious line to generate *ein2* and *wip1* homozygotes double mutants, and analyzed the expression of the stamina inhibitor *CmACS7*. Compared to *wip1* mutant, we found *wip1* and *ein2* homozygous double mutant fully hermaphroditic, despite the expression of the stamina inhibitor, *CmACS7* (Fig. 3c).

Ethylene has been associated with fruit shape development[26]. To investigate the role of *CmEIN2* in fruit shape we measured the ovary shape index (OSi) and the fruit shape index (FSi). The OSi is determined as the ratio of the ovary length by the ovary width. FSi is determined as the ratio of the fruit length by the fruit diameter at mature stage. We found that OSi and FSi are significantly higher in Charentais Mono line (WT) compared to *wip1/ein2* mutant. Conversely, the ovaries and fruits of *wip1/ein2* mutants were flatter, phenocopying the one of *acs7*[16] mutant, while the double mutants showed an additive effect (Fig. 3d–f). Collectively, our findings highlight the dual function of EIN2 in both sex determination and fruit shape. In lateral branches, upon the expression of *CmACS11*, EIN2 is recruited to repress the expression of the carpel inhibitor, *CmWIP1*. Subsequent to the expression of *CmACS7* in carpel primordia, EIN2 is recruited in the stamina inhibition processes. Following the sex determination phase, EIN2 promotes fruit shape elongation.

## EIN2 signaling plays contrasting roles in carpel development and stamina inhibition

To gain insight into the gene network associated with *CmEIN2* roles in stamina and carpel development, we examined the transcriptome of male, female, and hermaphrodite flower buds at stage 6. Male and female flower buds were collected from *ein2* and *wip1* loss-of-function mutants, respectively. Hermaphrodite flower buds were collected from *wip1* and *ein2* double mutant. Pairwise comparisons of the transcriptomes of male versus female, male versus hermaphrodite, and female versus hermaphrodite flowers, identified 8343, 5492 and 491 differentially expressed genes (DEG), respectively (Supplementary Fig. S6a, Supplementary Data 1). We grouped the differentially expressed genes by their expression patterns into 4 clusters: stamen suppression (S−), stamen development (S+), carpel suppression (C−), and carpel development (C+). Cluster S− and Cluster S+ denote genes that are upregulated, and genes that are downregulated, respectively, in female flowers compared to staminate flowers. Conversely, Cluster C− and Cluster C+ denote genes that are upregulated, and genes that are downregulated, respectively, in male flowers compared to pistillate

flowers (Fig. 3g, Supplementary Fig. S6b). Gene Ontology (GO) term enrichment analysis revealed cluster S− enriched in GO terms of biological process related to response to ethylene and negative regulation of signaling (Fig. 3g). Conversely, cluster S+ is enriched in GO terms associated to stamen formation, reproductive shoot system development, and positive regulation of transcription (Fig. 3g, Supplementary Fig. S7). Consistent with this, RT-qPCR analysis showed the upregulation of *ethylene responsive factor 1* (*ERF1*) and *EIN3 Binding F-Box Protein* (*EBF1*) associated with response to ethylene (GO: 0009723), and the downregulation of *PISTILLATA* (*PI*) and *bHLH91* related to stamen development (GO: 0048443), in female, compared to staminate flowers (Supplementary Fig. S6c, d). Cluster C+ was found enriched in GO terms related to regulation of histone acetylation, gametophyte development, and mitotic cell cycle, while cluster C- is enriched in GO terms related to response to abscisic acid, water deprivation, and protein dephosphorylation (Fig. 3g, Supplementary Fig. S7). Alongside, RT-qPCR analysis validated the downregulation of the carpel marker genes, *CRC* and *SEEDSTICK* (*STK*) (GO: 0035065), and the upregulation of abscisic acid (ABA) associated genes, *ABA RESPONSIVE ELEMENTS-BINDING FACTOR 3 (ABF3)* and Cytochrome P450 (*CYP707A10*) (GO: 0009737) in male, compared to pistillate flowers (Supplementary Fig. S6e, f). In summary, our RNA-seq analysis reveals two contrasting indirect roles of Cm*EIN2* in carpel development, and in stamina inhibition. In developing carpel, through inhibition of the carpel repressor, *CmWIP1* (Fig. 3b), EIN2 promotes the expression of genes associated with cell division and female gametophyte development, including the carpel identity gene, *CRC* (Fig. 3g, Supplementary Fig. S6e). In inhibited stamina, EIN2 promotes the expression of genes associated with negative regulation and signaling, and the repression of the stamina-promoting gene, *PI* (Fig. 3g, Supplementary Fig. S6c).

## Transposons of the *PIF/harbinger* family are highly active transposons in melon

To determine if *AndroPIF*, and related *PIF/Harbinger* elements, are active transposons in the melon genome, we analyzed two biological replicates of pooled genomic DNA of 50 CAM_WT plants, including CAM106 control line. To identify *AndroPIF* insertions, we performed transposon display followed by illumina sequencing (TEDseq[27]; Supplementary Fig. S8). We detected an average of 34 read coverage per *PIF/Harbinger* insertion loci (PIL), in both replicates, indicating the high sensitivity of TEDseq (Supplementary Fig. S9a). In total, we found 332

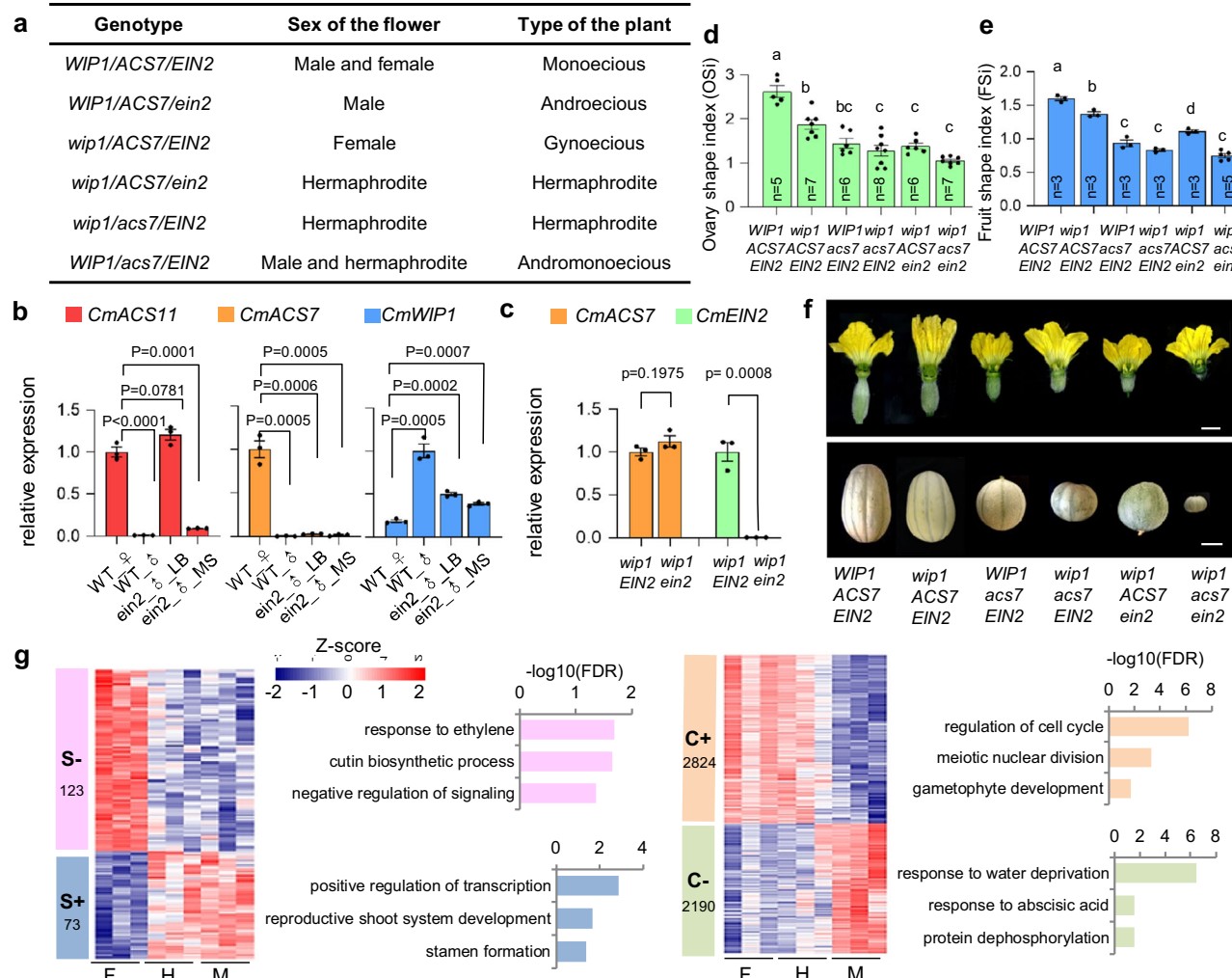

**Fig. 3 | CmEIN2 is required both for carpel development and stamina inhibition.**
**a** Sexual morphs of melon plants across combinations of alleles at different sex loci. **b** qPCR analysis of *CmACS11*, *CmACS7* and *CmWIP1* in Mono_WT female(♀) or male(♂) flowers, and in male flowers (♂) of CAM106(*ein2*) collected from main stems (MS), or lateral branches (LB). **c** qPCR analysis of *CmACS7* and *CmEIN2* in pistillate flowers of *wip1*,and *wip1/ein2* double mutants. P-values from an unpaired two-tailed Student's t-test are indicated in (**b**, **c**). Data in (**b**, **c**) are mean ± SE. (*n* = 3 biological replicates). **d**, **e** Ovary shape index(d) and fruit shape index(e) of different melon genotypes. Different letters indicate significant differences based on a one-way ANOVA with Tukey's multiple comparisons test. Data in (**d**, **e**) are mean ± SE. *n* is indicated for number of biological replicates in (**d**, **e**). **f** Ovary and fruit phenotypes of the genotypes shown in (**d** and **e**). Scale bar in the flower panel = 1 cm. Scale bar in the fruit panel = 5 cm. **g** Gene-wise hierarchical clustering heat map of differentially expressed genes (DEGs, adjusted *P* value < 0.05) between female (F, *wip1* mutant), hermaphrodite (H, *wip1/ein2* double mutant), and males (M, *ein2* mutant) flowers. DEGs (adjusted *P* value < 0.05) showing segregation into 4 clusters: stamen suppression (S−), stamen development (S+), carpel development (C+) and carpel suppression (C−). The z-score scale represents mean-subtracted regularized log-transformed read counts. GO terms specifying the biological process are shown at the right of the panels. Source data are provided as a Source Data file.

PILs, out of which 282 were common between CAM and Mono_WT, and 50 were specific to CAM (Fig. 4a, b; Supplementary Data 2). Further analysis of the 50 CAM PILs identified two insertions as ancient ones, validated by broken-paired reads in the CAM_WT parental line, resulting in 48 new *AndroPIF* insertions. These new insertions were distributed over the 12 melon chomosomes, with preferential insertion bias in gene rich regions (Fig. 4c). Furthermore, 83% of novel insertions locate in promoter regions, compared to 58% of the common ones (Fig. 4d). Local sequence analysis of AndroPIFs insertion sites revealed the consensus motif 'MWYTWARWK' (Fig. 4e), similar to that previously reported for PIF transposons[28,29]. Open chromatin, H3K9ac histone mark, and low CHG/CHH DNA methylation are key features associated with active chromatin[30]. We observed a distinct pattern wherein active chromatin marks exhibit a positive correlation with the insertion of new PILs (Fig. 4f, h). In contrast, common *AndroPIF* insertions, indicative of ancient insertions, are consistently located in

closed chromatin states, highlighting the temporal dynamics of chromatin from active to inactive, or reduced counter selection of insertions within inactive chromatin regions (Fig. 4g, h, Supplementary Fig. S9).

## *PIF/harbinger* SANT1 protein binds to PIF TIR sequences

AndroPIF, like most *PIF/Harbinger* transposons encodes two proteins: a nuclease and a DNA binding protein, SANT1 (Supplementary Fig. S2)[20]. Physical interaction between the two proteins is necessary for *PIF/Harbinger* transposition[28,31]. To further characterize *AndroPIF*, and related *PIF/Harbinger* transposon insertions in CAM melon line, we mapped genome wide DNA sequences binding to *AndroPIF* SANT1 protein, using amplified DNA affinity purification sequencing (amp-DAP-seq)[32]. A total of 1521 SANT1 binding peaks were detected in the CMiso reference genome (Supplementary Data 3). Of this, 60,4% (n = 919) overlapped sequences annotated as PIF, 21.5% (n = 327) were

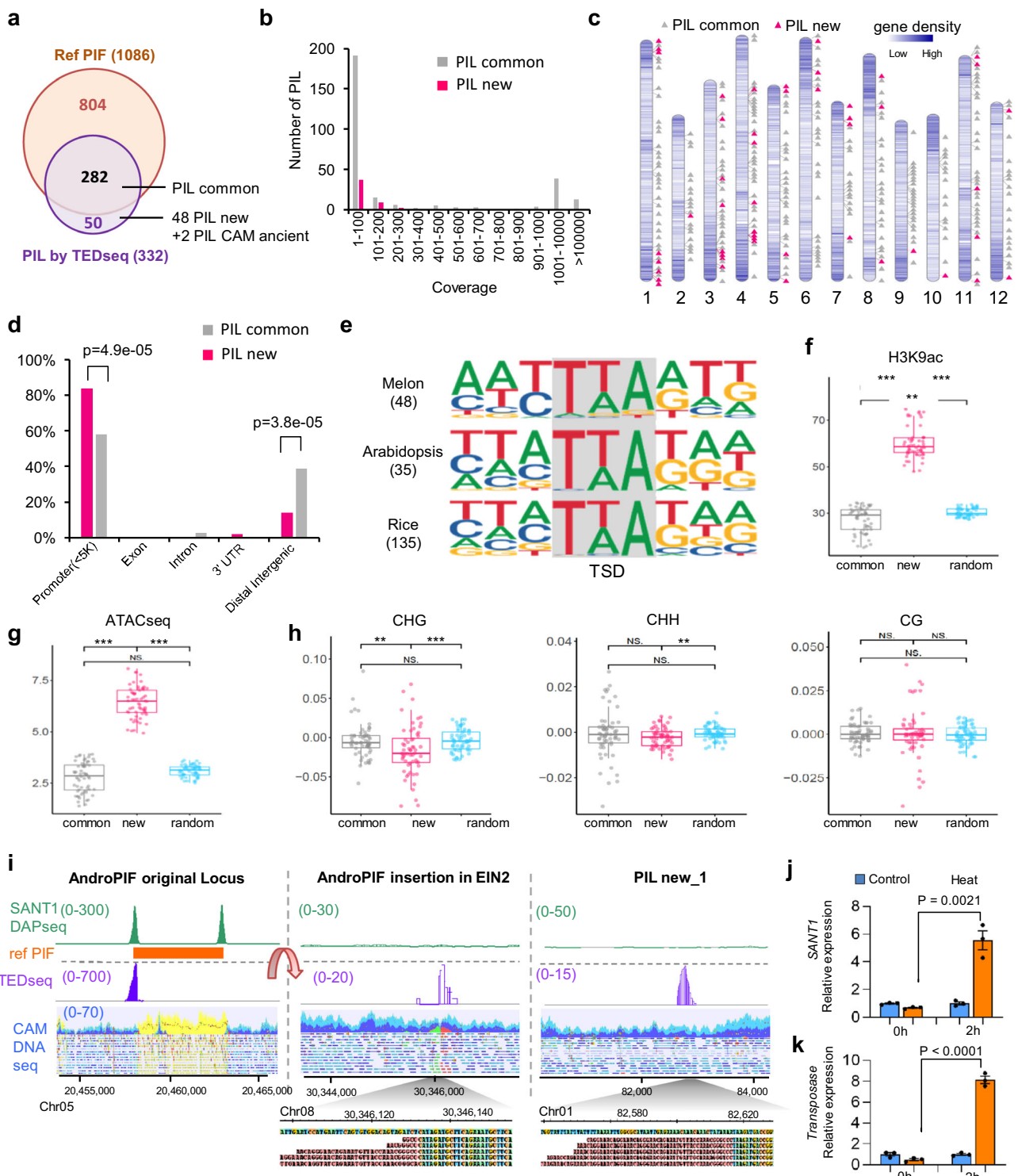

linked to sequences annotated as non-TE, and 18% bound sequences annotated as TE, but not-PIF (Supplementary Fig. S10a, b). Sequence analysis of SANT1 insertion sites in melon genome revealed two consensus motifs. Motif 1 'TTYYTGTTTCTTGTT' mapped at peak summits, while motif 2 'AGGGCCCGTTTGGTAACGTT' was located at the flanking sides of the summits (Supplementary Fig. S10c–d). Further sequence annotation identified part of motif 2 as the TIR sequence 'GGGCCCGTTTG' of *AndroPIF*, pointing to the role of SANT1 in binding to the transposon extremities. Consistent with this, we systematically observed two SANT1 ampDAP-seq peaks (double peaks, Supplementary Fig. S10e, f) mapping to loci annotated as PIF, including the

original copy of *AndroPIF* on chromosome 5, also captured by TEDseq (60,4% in Supplementary Fig. S10b). However, we found loci showing single peaks, corresponding to degenerate transposons that lost part of the TIR sequence, or random melon sequences that have similarities to TIR motif (18,5%, and 21.5% in Supplementary Fig. S10b). Several TEs have been shown to move following exposure to various environmental stressors[33,34], we analyzed whether the expression of *AndroPIF TRANSPOSASE* and *SANT1* genes are induced under heat stress conditions. CAM-WT plants were grown under control conditions (28 °C day/22 °C night), and shifted to heat stress conditions, for two hours at 42 °C. The expression of the *TRANSPOSASE* and *SANT1* genes were then

**Fig. 4 | *AndroPIF* transposon activity in melon genome. a** Venn plot of annotated *PIF/Harbinger* insertion Loci (PIL) in Mono genome (Ref PIL), and PIFs identified by TEDseq in poll of 50 CAM plants. **b** Sequence coverage of PILs identified by TEDseq. **c** Genomic position of PIL identified by TED-seq. PIL common, represent PILs that are common between CAM and Mono_WT. New PIL indicates new insertions observed in 50 plants descending from CAM parental line. **d** Distribution in percent of *AndroPIF* insertions in genes, and in intergenic regions. *P*-value from a Z-test are indicated and applied only to regions with non-zero values for both groups. **e** Target site insertion preference of new PILs, compared to *mPing* insertion sites, in Arabidopsis and rice[28,29]. **f–h**, Boxplots of H3K9ac (f), ATACseq(g), and DNA methylation(h) (CHG, CHH, and CG) levels in bin sizes of 100 bp in ±3 kb sequences flanking insertion sites of common PIFs, new PIFs, and random regions. For each boxplot, the lower and upper bounds of the box indicate the first (Q1) and third (Q3) quartiles, respectively, the center line indicates the median, and the whiskers extend to a maximum of 1.5 times the interquartile range (Q3–Q1). Number of bin is 60 for each box. Asterisks mark statistically significant differences (two-sided Wilcoxon rank-sum test, **$P < 0.01$, ***$P < 0.001$, NS. $P > 0.05$). **i** Genome browser view of SANT1 DAPseq peaks, TEDseq reads, and DNAseq, mapped on Mono genome. *AndroPIF* original locus in Mono genome, and in CAM_W (left), *AndroPIF* insertion in *EIN2* in CAM106 line (middle), and a new PIL in CAM_W progeny plant (right). Color scheme in DNAseq track: blue and purple, pair-end reads; yellow, highly repetitive reads; green and red, discordant reads. Soft-clipped reads are shown as sequences with red background. **J, k**, qPCR of SANT1 (**j**) and Transposase (**k**) of *AndroPIF* in young leaf of CAM-WT under heat stress (42 °C/ 32 °C) and control (28 °C/22 °C) conditions. *P*-values from an unpaired two-tailed Student's t-test are indicated. Data are mean ± SE. ($n$ = 3 biological replicates). Source data are provided as a Source Data file.

monitored by qPCR. We found the expression of both genes inducible by heat stress, pointing toward *AndroPIF* mobility triggered by abiotic factors (Fig. 4j, k).

### *PIF/Harbinger* display contrasted transposition in wild versus cultivated melons

Crop domestication involved recurrent selection of desirable traits, leading to a severe loss in sequence diversity. We investigated whether sequence diversity due to *PIF/Harbinger* transposon insertions has been also constrained in cultivated compared to wild melon accessions. Positions of *PIF/Harbinger* polymorphic insertions in the genomes of 480 cultivated (group CM, CA, and CAF in Fig. 5a) and 81 wild (group WM, WAF, and WA in Fig. 5a) accessions[35], collected from different geographic areas, was determined using short-read resequencing, and TEFLoN method[36,37]. Integration of PIF polymorphic insertion sites, SANT1 binding peak pairs, and search for TIR sequences resulted in the identification of 183 full-length *AndroPIF* copies (Supplementary Data 4, 5). The copies identified in the Mono-WT genome[38] are denoted as reference insertions, while others are labeled as non-reference insertions (see Supplementary Fig. S11). Phylogenetic tree construction based on the presence/absence of reference *AndroPIF* elements exhibits a clustering structure akin to that generated by SNP polymorphism (Fig. 5a, Supplementary Data 6)[35]. In contrast, the tree generated based on non-reference insertions displays a mixed clustering pattern, suggesting that transposition of *AndroPIF* elements contributed to increased diversity in these populations. We found most *PIF/Harbinger* reference insertions (169 out of 182) shared between cultivated and wild accessions (Fig. 5b), indicating that they are ancestral. In contrast, out of the 894 non-reference-insertion, 561 are found in single accessions, pointing toward recent transposition events (Fig. 5c). We call this new category of *PIF/Harbinger* transposons "non-reference-new-insertions". Distribution of non-reference-new-insertions in melon germplasms revealed active transpositions in wild melon accessions compared to melon breeding materials (Fig. 5d, e). As in CAM_WT, non-reference-new-insertions were overrepresented in promoter regions (65.14%), compared to reference insertion (57.09%), indicating that integration preference has shaped the genome landscape of *AndroPIF* insertions (Fig. 5f).

### Discussion

We investigated how new sex determination morphs could arise. During the propagation of an andromonoecious melon line, we encountered a spontaneous mutant, sexually converted from andromonoecy to androecy. This intriguing phenotype prompted us to study the causal mutation. We identified a *PIF/Harbinger* transposon as the origin of the sex transition, disrupting the expression of the *Ethylene Insensitive 2* (*CmEIN2*) gene. Through genetic and expression analyses, we unveiled a dual function of *CmEIN2* in sex determination and fruit shape. Upon the expression of *CmACS11* in carpel primordia, of flowers developing on lateral branches, CmEIN2 is recruited to

suppress the expression of the carpel inhibitor, *CmWIP1*. Subsequently, the expression of *CmACS7*, in developing carpel primordia, produce ethylene that is perceived in the stamina primordia, likely through a spatially differentially expressed ethylene receptors[17]. In stamina primordia, CmEIN2 is then recruited to mediate stamina inhibition, leading to female flower development. If *CmEIN2* is inactivated, the repression of *CmWIP1* expression is relieved leading to androecy. Inactivation of both *CmEIN2* and *CmWIP1* lead to hermaphrodite flower development (Fig. 6).

Transposons are powerful drivers of genome evolution and plasticity and contribute to genetic variability, potentially under stress conditions[34]. We thus tested whether *PIF/Harbinger* mobility is induced by adverse conditions. Structurally, *PIF/Harbinger* transposons are predicted to encode two proteins, a transposase and a DNA-binding protein. We found the expression of both genes activated by heat stress, pointing toward triggering of *PIF/Harbinger* mobility under adverse conditions. Similar stress-responsive transcription or movement has been reported in several TEs[33,39]. In rice, the DNA transposon mPing can be activated in response to cold and salt stress[27]; in tobacco, Tnt1 element can be induced by different biotic and abiotic stress[34].

Given the sensitivity of TEs to environmental threats, our findings suggest that androecy may arise as a result of transposition triggered by stress into *CmEIN2*. Ethylene is a key feminizing hormone in cucurbits[40,41]. Consistent with this scenario, we found *CmEIN2* required for the transcriptional repression of the male-promoting gene *CmWIP1*, leading to female flower development (Fig. 6). Under adverse conditions, fruit set could be costly, inactivation of *CmEIN2* relieves *CmWIP1* from the repression, and consequently leading to androecy. By producing only male flowers, androecious plants can have an advantage in promoting outcrossing.

The role of transposon mobilization in shaping sexual reproduction presented here could be a general mechanism. Besides *AndroPIF* transposon leading to androecy, we previously showed that the DNA transposon, *Gyno-hAT* is at the origin of male to female sex transition, resulting in gynoecious plants[16]. In species harboring sex chromosomes, TEs also tend to accumulate in genomic regions containing sex genes, underscoring their significant role in the emergence and development of sex chromosomes[42].

In addition to promoting female flower development, ethylene is required for inhibition of stamina in pistillate flowers (Fig. 6). We introduced *ein2* mutation in monoecious genetic background. We found plants heterozygous for *ein2* mutation having 20–40% of female flowers converted to bisexual, pointing to the quantitative role of ethylene in sex determination (Fig. 2d). Furthermore, ethylene is a key regulator for fruit elongation in melon. Consistent with this, fruits developing on plant harboring *ein2* mutation are significantly affected in both ovary and fruit shape indexes. In summary, our work proposes a model in which *CmEIN2* play a central role both in the control of unisexual flowers in monoecious plants as well as the emergence of male and female unisexual plants.

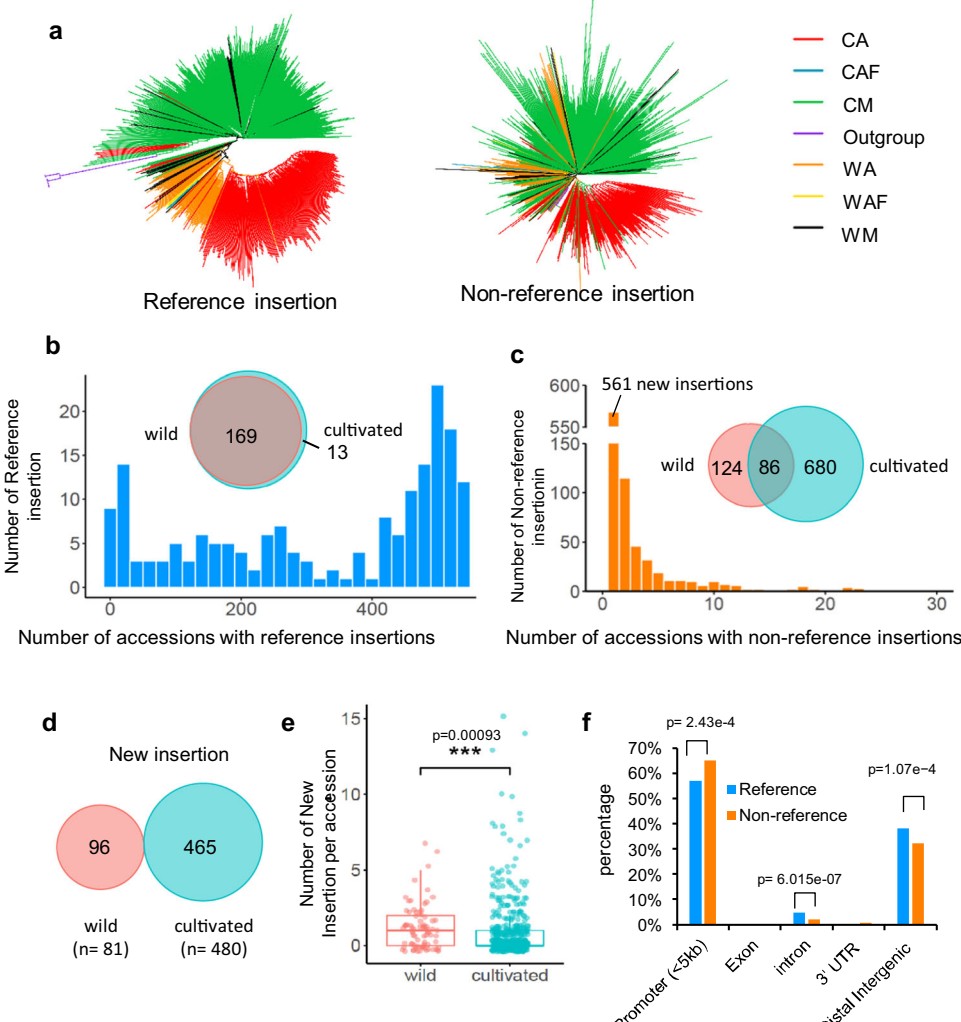

**Fig. 5 | _PIF_/_Harbinger_ transposition in wild and cultivated melon accessions.**
**a** Phylogenetic tree constructed using reference and non-reference _PIF_/_Harbinger_ insertion sites identified using genomic re-sequencing data of 567 melon accessions, and 7 outgroup relatives (Zhao, 2019). AF, African; WAF, wild African; CAF, cultivated African; WM, wild melo; CM, cultivated melo; WA, wild agrestis; CA, cultivated agrestis. **b**, **c** Histogram and venn plots of reference (**b**) and non-reference (**c**) _AndroPIF_ insertions identified in wild (WA and WM) and cultivated (CM and CA) accessions. **d** Venn plot of new insertions identified in wild and cultivated accessions. Number of accessions is indicated for each catalog. **e** Number of new-non-reference insertions per accession, in wild and cultivated accessions. _P_-values are calculated based on a two-sided Wilcoxon rank-sum test. For each boxplot, the lower and upper bounds of the box indicate the first (Q1) and third (Q3) quartiles, respectively, the center line indicates the median, and the whiskers extend to a maximum of 1.5 times the interquartile range (Q3–Q1). Number of accessions is indicated for each catalog in (**d**). **f** Distribution, in percents of _AndroPIF_ insertions in genes, and intergenic regions. _P_-values from an Z-test are indicated and applied only to regions with non-zero values for both groups.

The transcriptome analysis of flower buds collected from female (_wip1_/_EIN2_), hermaphrodite (_wip1_/_ein2_), and male (_ein2_) plants, revealed 4 clusters of gene expression patterns. We found S- cluster enriched in GO terms such as ethylene response genes, while S+ cluster contains several TFs, such as _PI_, implicated in stamen development. We also found C+ cluster enriched in GO terms related to female gametophyte development, and mitotic cell cycle, while C- cluster enriched in GO terms related to abscisic acid, water deprivation, and protein dephosphorylation. These results corroborate that ethylene signaling negatively regulates stamen development, while simultaneously exerting a positive effect on carpel development.

Annotating _PIF_/_Harbinger_ elements in plant genomes presents several challenges due to their high diversity in sequences, sizes, and organizations. Beside in silico annotation, we used SANT1 ampDAPseq analysis as a tool to identify and annotate _PIF_/_Harbinger_ transposons in melon genomes. In loci displaying ampDAPseq double peaks, we systematically found either a new or an ancient PIF element. However, in loci showing single ampDAPseq peaks, the accuracy of the method is weak, as the binding could be due to degenerate transposon or random sequences that has similarities to TIR sequences. Nevertheless, when we combined transposon annotation with SANT1 ampDAPseq analysis, the approach proved highly efficient in identifying new transpositions as well as ancient and degenerate transpositions.

Following the annotation of _PIF_/_Harbinger_ elements, we classified them into three categories, refence PILs, non-reference PILs and new PILs. We found most reference and non-reference PILs reflect the genetic relationship between cultivated and wild melon accessions, while new PILs corresponded to recent insertions. Interestingly, we found the new PILs more frequent in wild melon accessions compared to breeding materials, pointing toward their roles in plant adaptations. The movement of transposons in wild and cultivated plants is of particular interest because it can provide new genetic variation for plant breeding[43–45]. The majority of novel insertions are located in promoter regions. It will be interesting to explore their influence on the modulation of gene expressions and, as a result, on plant adaptation.

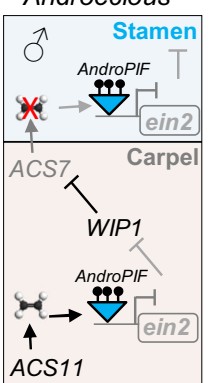
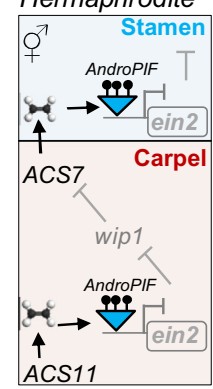

**Fig. 6 | Model of sex-determination pathway in melon integrating *EIN2* function.** The model integrates *ACS11*, *WIP1*, and *ACS7* sex-determination genes. Pink box indicates carpel primordia and blue box indicates stamen primordia. Spatial expression of *ACS11*, *ACS7*, or *WIP1* is restricted to carpel primordia. *EIN2* is expressed both in the carpel and stamina primordia. The chemical symbol indicates ethylene gas. Red cross indicates no ethylene produced by ACS7 or ACS11. Ethylene produced by ACS7 in the carpel is perceived in the stamina primordia likely through a spatially differentially expressed ethylene receptors[17]. Monoecious plants develop male and female flowers. *AndroPIF* insertion in *EIN2* leads to Androecy. *Ein2/wip1* double mutant is hermaphrodite. *EIN2*-mediated repression of *WIP1* depends on the expression of *ACS11*. *EIN2*-mediated repression of stamen requires the expression of *ACS7*.

Our work reveals a central role of *PIF/Harbinger* class of transposons in the emergence of sexual morphs in flowering plants. As *PIF/Harbinger* are present in organisms spanning the tree of life[19], our study will likely inspire future investigations beyond cucurbits.

## Methods

### Plant material
The CAM106 spontaneous mutant was discovered during the propagation of the andromonoecious line CAM_WT (Cucumis melo L. var. cantalupensis). To identify the causal mutation, we crossed CAM106 to CAM_Sister, an andromonoecious EMS-mutagenized mutant line derived from CAM_WT, and phenotype the F2 segregant plants. The plants were cultivated in a greenhouse with a daytime temperature of 27 °C, a nighttime temperature of 21 °C, and a photoperiod of 16 hours of light. The plants were assessed for flower sex type and fruit shape.

### Bulk segregant analysis
Genomic DNA was extracted from CAM106 X CAM_Sister F2 population. For bulk segregation analysis, equal amounts of genomic DNA from 10 individual plants with either andromonoecious or androecious phenotypes were pooled to create the andromonoecious and androecious pools. Subsequently, DNA libraries were prepared, using the Next Ultra DNA Library Prep Kit, and 150 bp paired-end reads sequenced on HiSeq 2000 Illumina platform. Reads were aligned to the reference Charentais Mono melon genome CMiso1.1[38] (here referred to as Mono_WT). The SNP-index across all loci was calculated, and the delta (Δ) SNP-index was determined by subtracting the SNP-indices of the two bulks at each locus (SNP-index_mutant – SNP-index_WT).

### Triple response assay
Melon seeds were germinated in a 10-litter wet box in the dark for 7 days at 27 °C, either with or without 50 µM ethylene. Ethylene solution at 50 µM concentration was prepared by diluting 1 M ethephon (Sigma) in 5 mM $Na_2HPO3$ buffer, as described previously[25].

### Sexual morph phenotyping and epistasis analysis
Monoecious melons carry only male flowers in the main stem, female flowers develop on the first nodes (L1) of lateral branches. The following flowers on L2 and L3 nodes of the lateral branches can be female depending upon plant age and environment. Melon also exhibits other sex morphs, andromonoecious, gynoecious, androecious and hermaphrodite. Androecious plants bear only male flowers. Gynoecious and hermaphrodite plants develop only female and hermaphrodite flowers, respectively. Andromonoecious plants develop separate male and hermaphrodite flowers. To position EIN2 function in the monoecy pathway we generated double mutants, through crossing of the single mutant and genotyping F2 segregant plants. The loss-of-function alleles leading to gynoecy (*wip1*), androecy (*acs11*) or andromomonoecy (*acs7*) used in the epistasis analysis are described in Boualem et al[14]., Martin et al[16]., and Boualem et al[15]., respectively[14–16]. Single, and double homozygous plants for *wip1*, *acs11*, *acs7*, and *ein2* mutations were phenotyped for sexual transition of the flowers on the male stem and lateral branches.

### Quantitative RT-PCR
Total RNA was extracted from frozen leaves or flowers. First-strand cDNA was synthesized from 2 pg of total RNA with the Superscript® III reverse transcriptase (Invitrogen). Primer design was performed using the Primer3 software (http://frodo.wi.mit.edu/cgi-bin/primer3/primer3_www.cgi). Primer sequences used are listed in Supplementary Table 2. Polymerase chain reactions were performed with the Bio-Rad CFX96 Real-time PCR apparatus, with qPCR MasterMix Plus for SYBR® Green I (Eurogentec, France). Gene expression is normalized to the expression levels of housekeeping genes: *CmActin2*. The qRT-PCR results were analyzed using a ΔΔCt methodology.

### RNA-seq assay
Flower buds at developmental stage 6 were sampled, and total RNA was extracted using the NucleoSpin RNA Plus kit (Macherey Nagel). The quality and concentration of the RNA were determined using Agilent RNA chips (Bioanalyser 2100). Three biological replicates for female (*wip1*), hermaphrodite (*wip1/ein2*), and male (*ein2*) flowers were used for library preparation. RNA-seq was performed on the Illumina NovaSeq 6000 with 150 bp paired-end reads. The reads were aligned to the reference genome CMiso1.1 by STAR (v2.7.5c)[46], and the mapped reads were assigned to genes with featureCount (v2.0.3)[47]. Differentially expressed gene calling was conducted using DiCoExpress[48], with a p-value adjusted <0.05. GO enrichments were completed by ClusterProfiler[49] with customized GO annotation.

### Whole-genome bisulfite sequencing
Genomic DNA of young leaves of CAM_WT and CAM106 were extracted with CTAB. The gDNA was used to generate the libraries and treated by bisulfite. Sequencing was done on Illumina NovaSeq 6000 with 150 bp paired-end reads. The reads were analyzed by MethylStar

pipeline[50]. The differentially methylated sites (DMRs) were identified by methylkit[51] and DMRcaller[52].

## TE display sequencing

TEDseq was performed on a DNA pool of 50 CAM_WT seedlings, and CAM-106 DNA, used a control. Genomic DNA was extracted using the DNeasy Plant Mini Kit (QIAGEN). Sequencing libraries were prepared from 1 μg of pooled DNA using the NEBNext® Ultra™ II DNA Library Prep Kit. PCR enrichment was carried out for 20 cycles, using Illumina P7 Primer with index, and P5 primers PIF-P1 or PIF-P2 (Supplementary Table 2; Supplementary fig. S8). Nested PCR enrichment was then carried out for 12 cycles, using PIF-P3 or PIF-P4. This process was independently repeated twice. Pair-end sequencing was carried out using the Illumina Miseq, generating 150 bp reads. To identify new TE insertions, soft-clipped and discordant reads that partially mapped to the reference sequence of AndroPIF were extracted. The reads were subsequently remapped to the CMisoV1.1 reference genome using Bowtie2 v2.3.5 with the arguments 'very-sensitive-local'[53]. Putative insertions were retained if they were supported by more than one set of 3 soft-clipped and discordant reads in at least one repeat at the same locus. TE target site analysis was performed with a Rscript, using packages 'dplyr', 'readxl', 'knitr', 'ggplot2', 'ggseqlogo' and 'Biostrings'. For each identified region, reads overlapping putative site and TE sequence were retrieved and trimmed with Cutadapt[54], and were aligned with Bowtie2 on CMisoV1.1 genome. The exact positions of the insertion sites were calculated by strand specificity and CICAR flag. A sequence of 6 nucleotides flanking the insertion site were extracted to calculate consensus matrix and motifs. The level of epigenetic marks around the TE insertion sites were calculated by Deeptools3.5.5 using the data described previously[17,55,56] (accession number: PRJNA383830).

## DNA affinity purification sequencing

*AndroPIF* SANT1coding sequence was isolated from leaf RNA by RT-PCR and cloned into pIX-HALO by Gateway recombination according to the manufacturer's protocol (Invitrogen). The primers used in cloning of SANT1 are listed in the Supplementary Table 2. Expression of HALO-SANT1 fusion protein and enrichment of DNA targets were performed as described previously[32]. The DNA used for DNA library preparations was extracted from the Charentais Mono line (WT). AmpDAP-seq was employed, utilizing amplification to eliminate DNA methylation. Library preparations followed the manufacturer's protocol for the NEBNext® Ultra™ II DNA Library Prep Kit for Illumina (Illumina). The final enriched DNA library was sequenced on a NextSeq500 as 2×75 nt reads. The control HALO protein AmpDAP-seq data were generated previously[18] (accession numbers: SRX15631305, SRX15631306).

## Transposable elements insertion detection in melon germplasm

The genomic sequences of the 567 *C. melo* accessions, and the 7 related species (Supplementary Data 6) were retrieved from previous study[35]. The genome wide identification of *AndroPIF* related elements was performed using the TEFLoN method and the McClintock pipeline[36].

TEs were subsequently annotated manually, using SANT1 ampDAPseq peaks, and the search of TIR sequences 'GGGCCCGTTTG', allowing 2 bp mismatches. CMisoV1.1 genome was used a reference genome for the mapping of the germplasm sequences.

The gff3 results from McClintock pipeline were converted to vcf format with the code present(REF0/0)/deletion(1/1) for reference insertion, and absence(0/0)/insertion(1/1) for non-reference insertion. For phylogenetic tree building, Euclidean distance was computed for bitwise matrix and the Neighbor Joining method was used for clustering. R packages 'vcfR', 'poppr', 'ggtree', and 'dplyr' were used for the analysis.

## Heat stress assays

Germinated seeds were grown for 3 weeks in the greenhousse at 27 °C daytime, 21 °C nighttime, and a photoperiod of 16 hours of light. For the heat stress assay, the plants were transferred to a growth chamber with a temperature set at 42 °C. Total RNA samples were extracted at 0 hours, and 2 hours after the initiation of the heat stress.

## Statistical analyses

All experiments were carried out in at least three biological replicates. Differences in mean for RT-qPCR data were assessed using an unpaired two-sided Student's t-test. For genomic distribution in percent of TE insertions in promoter, genes, or in intergenic regions, a two-sided Wilcoxon rank-sum test was performed and applied only to regions with non-zero values for both groups. For each boxplot, the lower and upper bounds of the box indicate the first (Q1) and third (Q3) quartiles, respectively, the center line indicates the median, and the whiskers extend to a maximum of 1.5 times the interquartile range (Q3–Q1). All the specific statistic methods were described in the figure legends.

## Reporting summary

Further information on research design is available in the Nature Portfolio Reporting Summary linked to this article.

## Data availability

Sequencing data generated in this study have been deposited in the NCBI Sequence Read Archive database under the accession number PRJNA1085522, including RNA-Seq, Bisulfite-Seq, TEDseq, and AmpDAP-seq. Previously published data used in this study are SRX15631305, SRX15631306, and PRJNA383830. Source data are provided with this paper.

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

## Acknowledgements

The authors thank Olivier Martin for the critical reading of the manuscript. We thank Vincent Rittener for technical assistance, and IPS2 glasshouse staffs and the research facilities provided by GAFL and the experimental unit, for plant handling. We thank the IPS2 bioinformatics team for providing help and computing resources. This work was supported by European Research Council(grant ERC-NectarGland Project, No.101095736 to A.Be.), Agence National de la Recherche (grant NECTAR, No. ANR-19-CE20-0023 to A.Be.), Inititiative d'Excellence Paris-Saclay (grant Lidex-3P, No. ANR-11-IDEX-0003-02 to A.Be. and A.Bo.).

## Author contributions

The conceptualization of the project was undertaken by A.Be. and A.Bo. The methodology. involved the efforts of H.H., S.Z., F.A.C., F.T., C.P., H.S.P., C.T., F.S., Q.C., and F.M. The investigation was conducted by A.Be., H.H., S.Z., F.A.C., M.V., A.Bo., C.D., and L.Q. The visualization was handled by H.H., S.Z., and M.V. Funding was acquired by A.Be., and project administration was led by A.Be. Supervision of the project was provided by A.Be., A.Bo., and C.D. The original draft of the writing was done by A.Be. and S.Z., and the review and editing were carried out by A.Be., S.Z., A.Bo., L.Q., and C.D.

## Competing interests

The authors declare no competing interests.
