## [Peer Review File · Nature Communications]

Harbinger transposon insertion in ethylene signaling gene leads to emergence of new sexual forms in cucurbitsREVIEWER COMMENTS

Reviewer #1 (Remarks to the Author):

Key results: Huang, Zhang, Choucha et al. identified a spontaneous mutant of sexual morphs in *Cucumis melo*, mapped the mutation to a DNA transposon insertion in the ethylene response gene *EIN2*, and show the role of *EIN2* in promoting female flower development. These results alone are significant, building up on a body of literature that increasingly shows the importance of transposon-induced polymorphism in population dynamics, and is particularly interesting in the context of sexual differentiation. The authors further show that the transposon responsible for the mutation, termed AndroPIF, is active in cultivated melons and their wild counterparts, is upregulated by heat stress and inserts preferentially into gene promoters.

Validity: Overall, the study is sound in its interpretation of the data: the genetics are solid, *EIN2*'s role in sexual determination and the characterization of AndroPIF activity are convincing. However, the transcriptomics suffers from flaws that preclude the publication of the manuscript in its current form. The description and representation of the data from the transposon-display and DAP-seq experiments would benefit from a rework and clarification.

Significance: The study is highly significant, showing the role of transposon-induced polymorphism in balancing sexual determination, which provides a notable counterpoint to a previous study by the authors (Martin et al. 2009 Nature). They extend their findings by characterizing the activity of AndroPIF in melon populations and its insertion preferences, which could open the way for future work on AndroPIF-mediated natural variation and its association with local adaptation.

Data & methodology: The genetics are carefully conducted to convincingly show that the AndroPIF insertion in *EIN2* 5' UTR is responsible for androecy. The phenotype of *ein2* mutants is clearly described and supports the authors' conclusions. The demonstration of AndroPIF activity, the characterization of its insertion preferences and population distribution is solid, but the presentation of the data could be improved. Most importantly, the experimental design of the transcriptome is flawed and should be revised.

Analytical approach: Statistical tests are appropriately used when indicated, but most tests are not described in the legends, so I cannot judge this aspect with certainty.

Suggested improvements: The transcriptomics should be removed, as they do not support

the authors' conclusions and are not central to the study. The presentation of the data in fig 4 and the results section should be clarified.

Clarity and context: Overall, the study and the authors' interpretations are clear. The text itself is accessible, but the overuse of jargon and abbreviations in lines 209 to 277 (and associated figures) makes the reading difficult to a reader not familiar with melon cultivars and genomes.

References: the manuscript refers to the existing literature appropriately.

My expertise: forward genetics, functional genomics, transcriptomics, epigenomics, transposons, DNA methylation, chromatin.

I recommend the editor to invite the authors to revise their manuscript before acceptance.

Major points:

1) The transcriptomics experimental design is flawed.

The role of EIN2 in regulating carpel development and stamina inhibition is supported by the analysis of *ein2* mutant phenotypes and is consistent with the known role of ethylene in female flower development. However, the transcriptome study suffers from the lack of WT controls, which precludes any interpretation of the function of EIN2 or WIP1 in regulating gene expression.

The authors conclude:

- "In developing carpel, EIN2 promotes the expression of genes associated with cell division and gametophyte development." This is not supported by the data: genes that promote carpel development (fig 3g, C+ cluster) are highly expressed in hermaphrodite flowers, where EIN2 is knocked out.

The authors compared *ein2* male flowers with *wip1* female flowers and *ein2 wip1* hermaphrodite flowers. Where are the WT controls for male and female flowers? In a typical controlled experiment, only one parameter varies to allow for comparisons. What conclusions can be drawn by comparing different cell types originating from distinct mutants?

I understand why the authors chose this approach: each mutant only produces one type of flowers. The authors cannot simply compare male flowers of *ein2* and *wip1* because there are no male flowers in *wip1* mutants.

The heatmap presented in fig 3g allows for the following conclusions:

- Transcriptomes of female and male flowers are essentially opposite.
- Transcriptomes of hermaphrodite flowers are intermediate between female and male flowers.
- Female, male, and hermaphrodite flowers have specific transcriptome signatures, with down and upregulated genes that are enriched for particular GO terms.

No conclusions can be drawn about the roles of EIN2 and WIP1 in controlling gene expression programs without WT controls.

Of note, however: in the discussion, the authors observe that S- cluster are enriched for ethylene response genes, corroborating the notion that ethylene negatively regulates stamen development. This conclusion about the function of ethylene, rather than EIN2, is indeed supported by the transcriptome data, but it is not novel (see review by Li et al. 2019 Front. Plant Sci. <https://doi.org/10.3389/fpls.2019.01231>).

I see two possible alternatives:

- Compare WT flowers with mutant flowers of the same sex.
 - o Compare wip1 to WT female flowers.
 - o Compare ein2 to WT male flowers.
 - o Compare wip1 ein2 to a mix of WT male and female flower transcriptome, but this will be very approximative.
- Compare floral buds at an early developmental stage, prior to sexual differentiation.
 - o Determine WIP1 and EIN2 expression during floral development (before sexual differentiation) to find a developmental window where both genes are expressed, meaning wip1 and ein2 mutations are likely to provoke transcriptome changes. Depending on the expression pattern, one might have to choose a different stage for each mutation.
 - o Analyze transcriptomes of WT, wip1, ein2 and wip1 ein2 at the same stage, to allow for comparisons. This should tell authors which genes expressed early in floral differentiation are important for sex determination, and which ones are controlled by EIN2 & WIP1.

The second alternative should be more interesting but is more challenging.

Given the relatively minor importance of the transcriptome data in the current manuscript, I suggest the authors remove the experiment entirely.

2) Figure 4 and associated text need some rework.

The results themselves are convincing, but the data representation, figure legends and

result descriptions are unclear and hard to follow. The overuse of jargon makes reading difficult for the reader.

- The order of figure 4 panels is scrambled relative to the text, figure 4d is referred first after fig 4i.
- I would suggest removing SANT1 peaks from fig 4a. Instead, show which PILs are common, which are new, this will ease the understanding of figure 4b and 4e.
- Legend for fig4b should explain what is “common” and “new”.
- “CK” is unnecessary jargon, “random” would be just fine.
- Fig 4f: please show the actual motif sequence logo, rather than frequencies, to compare motifs across species, to improve readability.
- Line 224: “exhibit a positive correlation with the insertion of new PILs.”, please refer to the figure here.
- From which line was extracted the DNA used for DAP-seq? Was the DNA amplified to remove DNA methylation? These details are important and should appear in the methods and results.
- Fig S10A: could you show the reproducibility by using average score (RPM or other) per bin? The raw data is too heavily transformed to appropriately compare replicates with the current method.
- 23 % + 19% of DAP-seq peaks are found outside of PIF elements. The authors claim that their method can map PIF transposons in genomes (line 246), but they have not shown if these 42% of peaks outside of PIF elements were un-annotated PIF elements or false positives. 42% of non-PIF loci is a lot. The authors should remove their claim or analyze the sequences of these peaks to show that they are PIF elements.
- Colors of different groups in fig 5a are too similar, making it hard to read.

Minor points:

- Line 65: please include a reference (a review will do here)
- Most statistics are not described in the figure legends: Fig 1g, 3b, 3d, 3e, 4g-i
- Authors comment differences for which no p-value is reported: In fig 3b, the authors interpret that the increase in WIP1 expression in CAM106-m-LB relative to Mono-F is responsible for androecy. Yet, no p-value is reported to compare the data of these two groups.

- In fig 5f, the proportion difference at promoters between the two groups is not supported by a statistical test, and the difference is weak regardless, meaning the biological interpretation made by the authors should be more cautious (lines 276-277).
- The discussion emphasizes the AndroPIF stress response too much, given that this aspect has been only superficially investigated by the authors. Is there any ecological data associated with the wild accession habitats? Could one correlate the number of new AndroPIF insertions with elevated temperatures of the ecosystem?

Pierre Bourguet

Reviewer #2 (Remarks to the Author):

This MS reports gene identification of a melon mutant showing a transition from bisexual flower to unisexual male flower. The authors found that a Harbinger transposon insertion disrupted the CmEIN2 gene, which is involved in ethylene signaling and female flower development. Further single, double and even triple mutant analysis, together with the transcriptome analysis, revealed a dual role of CmEIN2 in both sex determination and fruit shape formation. Genome-wide analysis found that the transposon mobilization is preferentially integrated in the promoter regions and triggered by heat stress. Distribution of the transposon in a large collection of melon germplasms is also analyzed and the active transposition happens more in wild accessions than that in cultivated accessions. This study provides novel materials and insights into the regulation of sex determination in melon through beautiful genetic experiments, and should facilitate melon breeding in practice. The following points may be addressed further.

- 1, Line 161, please describe in a bit detail how the single and double mutant of ein2 and wip1 are generated.
- 2, Fig 3b, can the genotype of each material be indicated according to 3a? So readers may easily know the relationship.
- 3, Lines 218-220, the authors identified 48 new insertion site (PILs). And 83% of the insertions are located in the promoter regions. Can the authors test a few of these cases to

see if these insertions would affect the expression of the neighbor genes?

4, Line 252-253, the sentence may be revised. One 'triggered' may be removed. Please check.

5, In Fig. 4, the last panel should be 'l' but not 'i'. Please correct the figure panel, and the legends.

6, It would be great if the descriptions in the text followed the order of the panels in Figure 4.

7, Fig 5a, the categories of 'CA, CAF ...' should be explained a bit more detail. Fig 5d, the numbers of wild and cultivated accessions should be given on the panel.

8, Fig 6, does the 'ACS7' move from the carpel to the stamen? If yes, please explain a bit more. If not, can the 'ACS7' be included only in the stamen organ? What do the authors mean by 'blue box indicates stamen...'? I did not see the blue box. Please explain a bit more for each flower type about the regulatory pathway for carpel and stamen development in figure legends.

9, Fig 3c, how about the ACS11 expression levels in wip1 ein2 double and single mutant? It would be great if this gene expression is examined to check the model.

10, In the introduction part, the CmWIP1 may be explained in a bit more detail, e.g., encoding a C2H2 zinc finger-type transcription factor?

11, In the discussion part, since the MS is more focused on ethylene signaling components and their regulation on carpel and stamen development, I would suggest that the authors explained the model in the first paragraph. The transposon could be discussed in later part.

12, In abstract, line 21, 'fruit shape' is better changed to 'fruit shape formation'.

Reviewer #3 (Remarks to the Author):

This manuscript describes an extensive, comprehensive, and well-performed set of experiments to identify the genetic basis for a spontaneous novel mutation causing conversion to monoecy from androecy in melon. QTL-seq followed by fine-mapping identified a transposon insertion in an ethylene signal transduction gene, EIN2. Ethylene is a known key player in sex expression in cucurbits. The role of EIN2 in the observed phenotypes was verified by identifying and characterizing other EIN2 mutants; and the

effect of the mutant on known sex expression genes in melon as well as overall gene expression changes was characterized. The authors then characterized the distribution of this transposon (Harbinger) in the melon genome, identifying both ancient and recent transpositions. Finally, they examined distribution of recent and historical transpositions in cultivated and wild accessions, indicating how transposon activity can be an evolutionary contributor to variation in sex types.

I only have minor suggestions:

1. Please provide references for the sex phenotype description lines 66-71
2. Line 205. "EIN2 promotes the expression of genes associated with cell division and gametophyte development." It seems more likely that this is an indirect effect of promoting carpel development, which in turn leads to the cell division and, especially, gametophyte development rather than a direct effect of EIN2.
3. Fig 4 questions:
 - a. Fig 4 a refers to RefPIF, SANT1, and PIFs identified by TEDseq; however the numbers included within the RefPIF circle do not equal the total provided
 - b. Line 220. "Furthermore, 83% of novel insertions locate in promoter regions, compared to 58% of the common ones." Should this reference Fig 4e?
 - c. The labeling for Fig 4d is confusing. Does 'single peak' refer to the TIR sequence and 'double peak' to the SANTI sites? If so, it might be helpful to label as those instead. The fact that peaks are single and double is evident from the graph.
 - d. Why are there not SANTI peaks for the new insertions shown in Fig 4j?

Minor grammar notes:

Line 73. 'encodes for' should either be 'encodes' or 'codes for'

Line 91. 'buckets of flowers' should be 'clusters of flowers'

Line 92. Write out CRC at first mention

Line 124. Delete 'of which'

Line 214. 'sensibility' should be 'sensitivity'

Line 258. 'constraint' should be 'constrained'

Line 295. 'threads' should be 'threats'

REVIEWER COMMENTS:

We have highlighted in black ink the comments of the reviewers. Our answers or comments to the points raised by the reviewers are typed in red ink.

Reviewer #1 (Remarks to the Author):

Key results: Huang, Zhang, Choucha et al. identified a spontaneous mutant of sexual morphs in *Cucumis melo*, mapped the mutation to a DNA transposon insertion in the ethylene response gene EIN2, and show the role of EIN2 in promoting female flower development. These results alone are significant, building up on a body of literature that increasingly shows the importance of transposon-induced polymorphism in population dynamics, and is particularly interesting in the context of sexual differentiation. The authors further show that the transposon responsible for the mutation, termed AndroPIF, is active in cultivated melons and their wild counterparts, is upregulated by heat stress and inserts preferentially into gene promoters.

Validity: Overall, the study is sound in its interpretation of the data: the genetics are solid, EIN2's role in sexual determination and the characterization of AndroPIF activity are convincing. However, the transcriptomics suffers from flaws that preclude the publication of the manuscript in its current form. The description and representation of the data from the transposon-display and DAP-seq experiments would benefit from a rework and clarification.

Significance: The study is highly significant, showing the role of transposon-induced polymorphism in balancing sexual determination, which provides a notable counterpoint to a previous study by the authors (Martin et al. 2009 Nature). They extend their findings by characterizing the activity of AndroPIF in melon populations and its insertion preferences, which could open the way for future work on AndroPIF-mediated natural variation and its association with local adaptation.

Data & methodology: The genetics are carefully conducted to convincingly show that the AndroPIF insertion in EIN2 5' UTR is responsible for androecy. The phenotype of *ein2* mutants is clearly described and supports the authors' conclusions. The demonstration of AndroPIF activity, the characterization of its insertion preferences and population distribution is solid, but the presentation of the data could be improved. Most importantly, the experimental design of the transcriptome is flawed and should be revised.

Analytical approach: Statistical tests are appropriately used when indicated, but most tests are not described in the legends, so I cannot judge this aspect with certainty.

Suggested improvements: The transcriptomics should be removed, as they do not support the authors' conclusions and are not central to the study. The presentation of the data in fig 4 and the results section should be clarified.

Clarity and context: Overall, the study and the authors' interpretations are clear. The text itself is accessible, but the overuse of jargon and abbreviations in lines 209 to 277 (and associated figures) makes the reading difficult to a reader not familiar with melon cultivars and genomes.

References: the manuscript refers to the existing literature appropriately.

My expertise: forward genetics, functional genomics, transcriptomics, epigenomics, transposons, DNA methylation, chromatin.

We thank the reviewer for highlighting the quality of our work.

I recommend the editor to invite the authors to revise their manuscript before acceptance.

Major points:

1) The transcriptomics experimental design is flawed.

The role of EIN2 in regulating carpel development and stamina inhibition is supported by the analysis of *ein2* mutant phenotypes and is consistent with the known role of ethylene in female flower development. However, the transcriptome study suffers from the lack of WT controls, which precludes any interpretation of the function of EIN2 or WIP1 in regulating gene expression.

The authors conclude:

- "In developing carpel, EIN2 promotes the expression of genes associated with cell division and gametophyte development." This is not supported by the data: genes that promote carpel development (fig 3g, C+ cluster) are highly expressed in hermaphrodite flowers, where EIN2 is knocked out.

The authors compared ein2 male flowers with wip1 female flowers and ein2 wip1 hermaphrodite flowers. Where are the WT controls for male and female flowers? In a typical controlled experiment, only one parameter varies to allow for comparisons. What conclusions can be drawn by comparing different cell types originating from distinct mutants?

I understand why the authors chose this approach: each mutant only produces one type of flowers. The authors cannot simply compare male flowers of ein2 and wip1 because there are no male flowers in wip1 mutants.

The heatmap presented in fig 3g allows for the following conclusions:

- Transcriptomes of female and male flowers are essentially opposite.
- Transcriptomes of hermaphrodite flowers are intermediate between female and male flowers.
- Female, male, and hermaphrodite flowers have specific transcriptome signatures, with down and upregulated genes that are enriched for particular GO terms.

No conclusions can be drawn about the roles of EIN2 and WIP1 in controlling gene expression programs without WT controls.

Of note, however: in the discussion, the authors observe that S- cluster are enriched for ethylene response genes, corroborating the notion that ethylene negatively regulates stamen development. This conclusion about the function of ethylene, rather than EIN2, is indeed supported by the transcriptome data, but it is not novel (see review by Li et al. 2019 Front. Plant Sci. <https://doi.org/10.3389/fpls.2019.01231>).

I see two possible alternatives:

- Compare WT flowers with mutant flowers of the same sex.
 - o Compare wip1 to WT female flowers.
 - o Compare ein2 to WT male flowers.
 - o Compare wip1 ein2 to a mix of WT male and female flower transcriptome, but this will be very approximative.
- Compare floral buds at an early developmental stage, prior to sexual differentiation.
 - o Determine WIP1 and EIN2 expression during floral development (before sexual differentiation) to find a developmental window where both genes are expressed, meaning wip1 and ein2 mutations are likely to provoke transcriptome changes. Depending on the expression pattern, one might have to choose a different stage for each mutation.
 - o Analyze transcriptomes of WT, wip1, ein2 and wip1 ein2 at the same stage, to allow for comparisons. This should tell authors which genes expressed early in floral differentiation are important for sex determination, and which ones are controlled by EIN2 & WIP1.

The second alternative should be more interesting but is more challenging.

Given the relatively minor importance of the transcriptome data in the current manuscript, I suggest the authors remove the experiment entirely.

We fully agree with the reviewer that the transcriptomics data could be seen by a non-specialist as descriptive. We also believe that most of the questions of the reviewer regarding the design of the RNA seq are due to the description of the genetic material we used for RNAseq, that requires more explanation.

In the new version, and to comply with the reviewer critics, we inserted in material and method a paragraph that better explains the choice of the genotypes for RNAseq analysis. We also rewrote the RNAseq analysis section to make it more accessible to a non-specialist.

Our genetic analysis reveals that EIN2 play a dual role in carpel development, and stamina inhibition.

Upon expression of *CmACS11*, in carpel primordia of flowers developing in lateral branches, EIN2 is recruited to repress the expression of the carpel inhibitor, *CmWIP1*. Subsequently, *CmACS7* is expressed in carpel primordia to produce ethylene. This ethylene signal is perceived in the staminal primordia where EIN2 is recruited to mediate staminal inhibition, leading to female flower development. Following the sex determination phase, EIN2 promotes fruit shape elongation.

In *ein2* loss of function mutant, the plant develops only male flowers, because *wip1* repression is relieved. Thus, if we want to investigate the role of EIN2 in staminal inhibition we are forced to compare *wip1* mutant (female flowers) to *ein2/wip1* double mutant (hermaphrodite flowers), which we did.

In the transcriptomic analysis, we identified four gene clusters: stamen suppression (S-), stamen development (S+), carpel development (C+) and carpel suppression (C-). Consistent with the genetic model, in female/hermaphrodite flower comparison, we found EIN2 signaling associated with transcription inhibition of *PISTILLATA (PI)* and *bHLH91* related to stamen development (GO: 0048443).

In hermaphrodite versus male flower comparison, Cluster C+ (developing carpel) was enriched in GO terms related to regulation of histone acetylation, gametophyte development, and mitotic cell cycle, while cluster C- (inhibited carpel) revealed GO term enrichment for response to abscisic acid, water deprivation, and protein dephosphorylation (Fig. 3i). RT-qPCR analysis confirmed the upregulation of the carpel marker genes, *CRC* and *SEEDSTICK (STK)* (GO: 0035065), and the downregulation of abscisic acid (ABA) associated genes, *ABA RESPONSIVE ELEMENTS-BINDING FACTOR 3 (ABF3)* and Cytochrome P450 (*CYP707A10*) (GO: 0009737) in carpel-bearing flowers in comparison to males

In conclusion the choice of the genotypes permitted us to ease flowers sampling at stage 6, a stage where it is impossible to distinguish the sex of the flowers.

We hope that this explanation clarifies our choices of the genotypes for RNAseq analysis.

2) Figure 4 and associated text need some rework.

The results themselves are convincing, but the data representation, figure legends and result descriptions are unclear and hard to follow. The overuse of jargon makes reading difficult for the reader.

- The order of figure 4 panels is scrambled relative to the text, figure 4d is referred first after fig 4i.

Thank you for pointing this, we modified the labeling of the panels so that the descriptions in the text follow the order of the panels.

- I would suggest removing SANT1 peaks from fig 4a. Instead, show which PILs are common, which are new, this will ease the understanding of figure 4b and 4e.

Thank you for pointing this. We modified the panel as suggested by the reviewer.

- Legend for fig4b should explain what is "common" and "new".

Common and new explained in the figure legend and in the text file.

- "CK" is unnecessary jargon, "random" would be just fine.

CK replaced by random.

- Fig 4f: please show the actual motif sequence logo, rather than frequencies, to compare motifs across species, to improve readability.

As requested by the reviewer, we modified the panel accordingly.

- Line 224: "exhibit a positive correlation with the insertion of new PILs.", please refer to the figure here.

We inserted the figure "Fig4 f to h" in the text.

- From which line was extracted the DNA used for DAP-seq? Was the DNA amplified to remove DNA methylation? These details are important and should appear in the methods and results.

The DNA was extracted from CAM line. As transposon are highly methylated, to detect the PIF in the genome, we preferred to use amplified DNA library to eliminate the DNA methylation. For more clarity we specify in the materials and methods and the text that we used ampDAP-seq.

- Fig S10A: could you show the reproducibility by using average score (RPM or other) per bin? The raw data is too heavily transformed to appropriately compare replicates with the current method.

Thank you for the suggestion. We have generated a new panel (Fig S10A) using RPKM and bins of 10k. Spearman correlation coefficient is indicated on the plot.

- 23 % + 19% of DAP-seq peaks are found outside of PIF elements. The authors claim that their method can map PIF transposons in genomes (line 246), but they have not shown if these 42% of peaks outside of PIF elements were un-annotated PIF elements or false positives. 42% of non-PIF loci is a lot. The authors should remove their claim or analyze the sequences of these peaks to show that they are PIF elements.

Thank you for pointing this. DNA transposons of the P Instability Factor (PIF)/Harbinger superfamily are characterized by TIR sequences at their extremities. SANT1 DAP-seq analysis revealed the binding of SANT1 to both TIR sequences of PIFs, leading to double peaks. When we have double peaks, we are sure 100% that we are detecting a PIF (The two TIR sequences are spaced by a transposase and SANT1 protein).

When we detect only one peak, the situation is more complex, and we have two possibilities: Either we are in a situation of a degenerate PIF that lost one TIR, or random sequences that has similarities to TIR sequences.

For more clarity we re-annotated all the non-PIF sequences annotated as TE, and analysed further the non-TE bound sequences. We inserted the new panel, as well as the heat map of SANT1 peak density in the region ± 3 kb around all the reference PIFs in Figure S10. We also edited the text in the result section to explain better the DAPseq data.

In the discussion section, we inserted the following section highlighting the efficiency and the limitations of our method: <<Annotating *PIF/Harbinger* elements in plant genomes presents several challenges due to their high diversity in sequences, sizes, and organizations. Beside *in silico* annotation, we used SANT1 ampDAPseq analysis as a tool to identify and annotate *PIF/Harbinger* transposons in melon genomes. In loci displaying ampDAPseq double peaks, we systematically found either a new or an ancient PIF element. However, in loci showing single ampDAPseq peaks, the accuracy of the method is weak, as the binding could be due to degenerate transposon or random sequences that has similarities to TIR sequences. Nevertheless, when we combined transposon annotation with SANT1 ampDAPseq analysis, the approach proved highly efficient in identifying new transpositions as well as ancient and degenerate transpositions. >>

- Colors of different groups in fig 5a are too similar, making it hard to read.

We selected new contrasted colors for fig 5a.

Minor points:

- Line 65: please include a reference (a review will do here)

A review was included here.

- Most statistics are not described in the figure legends: Fig 1g, 3b, 3d, 3e, 4g-i

To avoid long legends, we have put experimental details in the Material and methods and short references to the statistical tools in the legends. For more clarity, and as requested by the reviewer, we have added more explanation on the figure legends on the statistical tests.

- Authors comment differences for which no p-value is reported: In fig 3b, the authors interpret that the increase in WIP1 expression in CAM106-m-LB relative to Mono-F is responsible for androecy. Yet, no p-value is reported to compare the data of these two groups.

As suggested by the reviewer, we inserted the exact P-value in all the panels.

- In fig 5f, the proportion difference at promoters between the two groups is not supported by a statistical test, and the difference is weak regardless, meaning the biological interpretation made by the authors should be more cautious (lines 276-277).

The proportion differences were significant between the two groups, performed by an unpaired Mann-Whitney U test at the promoter, intron, and distal intergenic regions, where contain non-zero values for both groups.

- The discussion emphasizes the AndroPIF stress response too much, given that this aspect has been only superficially investigated by the authors. Is there any ecological data associated with the wild accession habitats? Could one correlate the number of new AndroPIF insertions with elevated temperatures of the ecosystem?

Ecological studies that combine molecular genetics analysis and population genetics are instrumental to deeply understand the role of PIFs in adaptation.

Our analysis reveals that wild and cultivated materials are diverging for PIF genetic diversity, likely because of the intensive breeding that reduced PIF genetic pool, stabilizing the genomes of cultivated varieties. This finding extends previous reports stating that intense breeding reduces sequence diversity in coding genes, to transposable element diversity in cultivated genomes. To comply with the reviewer's request, we discussed further this point in the new version.

Pierre Bourguet

Reviewer #2 (Remarks to the Author):

This MS reports gene identification of a melon mutant showing a transition from bisexual flower to unisexual male flower. The authors found that a Harbinger transposon insertion disrupted the *CmEIN2* gene, which is involved in ethylene signaling and female flower development. Further single, double and even triple mutant analysis, together with the transcriptome analysis, revealed a dual role of *CmEIN2* in both sex determination and fruit shape formation. Genome-wide analysis found that the transposon mobilization is preferentially integrated in the promoter regions and triggered by heat stress. Distribution of the transposon in a large collection of melon germplasms is also analyzed and the active transposition happens more in wild accessions than that in cultivated accessions. This study provides novel materials and insights into the regulation of sex determination in melon through beautiful genetic experiments, and should facilitate melon breeding in practice. The following points may be addressed further.

We thank the reviewer for highlighting the quality of our work, and for the suggestions to improve the reading of our manuscript.

1, Line 161, please describe in a bit detail how the single and double mutant of *ein2* and *wip1* are generated.

For more clarity, and as requested by the reviewer, we inserted a new section in materials and methods describing the positioning of *EIN2* function in the monoecy pathway (Sexual morph phenotyping and epistasis analysis). We also describe better the crossing in the result part as follow: << In contrast, the repression of *CmWIP1* expression was relieved, leading to male plants (Fig. 3a-b). To investigate the role of *CmEIN2* in stamina inhibition, we crossed *ein2* mutant to *wip1* gynoecious line to generate *ein2* and *wip1* homozygotes double mutants. Compared to *wip1* mutant, we found *wip1* and *ein2* homozygous double mutants fully hermaphroditic, despite the expression of the stamina inhibitor, *CmACS7* (Fig. 3c).>>

2, Fig 3b, can the genotype of each material be indicated according to 3a? So readers may easily know the relationship.

Thank you for the suggestion, for more clarity, we inserted the genotypes in the panels.

3, Lines 218-220, the authors identified 48 new insertion site (PILs). And 83% of the insertions are located in the promoter regions. Can the authors test a few of these cases to see if these insertions would affect the expression of the neighbor genes?

In our work we analysed in details AndroPIF insertion in *ein2*, including the impact on the expression and the methylation. Concerning the other PILs, our analysis concentrated on their validation as new PILs.

The suggestion of the reviewer is indeed a valuable approach to assess the function of the new insertion events. Unfortunately, the plant material which was used to make TEDseq library was not kept for RNA-seq or qPCR experiment. Since each new insertion event occurred independently, testing the impact of insertions on gene expression will requires performing new TEDseq and qPCR/RNAseq simultaneously. This will delay significantly our publication. To take in consideration the reviewer's comment, we put in front in the discussion, the importance of studying the PIL impact on gene expression, by inserting the following paragraph << The majority of novel insertions are located in promoter regions. It will be interesting to explore their influence on the modulation of gene expressions and, as a result, on plant adaptation.>>

4, Line 252-253, the sentence may be revised. One 'triggered' may be removed. Please check.

Thank you for pointing this grammatical error. We removed the second triggered.

5, In Fig. 4, the last panel should be 'l' but not 'i'. Please correct the figure panel, and the legends.

Thank you for pointing this labeling error. We corrected it.

6, It would be great if the descriptions in the text followed the order of the panels in Figure 4.

Thank you for pointing this, we modified the labeling of the panels so that the descriptions in the text follow the order of the panels.

7, Fig 5a, the categories of 'CA, CAF ...' should be explained a bit more detail. Fig 5d, the numbers of wild and cultivated accessions should be given on the panel.

As requested by the reviewer, we provided more information on the panels and the figure legend.

8, Fig 6, does the 'ACS7' move from the carpel to the stamen? If yes, please explain a bit more. If not, can the 'ACS7' be included only in the stamen organ? What do the authors mean by 'blue box indicates stamen...'? I did not see the blue box. Please explain a bit more for each flower type about the regulatory pathway for carpel and stamen development in figure legends.

ACS7 do not move from carpel to stamina. The expression of ACS11, ACS7 or WIP1 is restricted to carpel primordia tissues. EIN2 is expressed both in the carpel and stamina primordia tissues. Ethylene produced by ACS7 in the carpel is likely perceived in the stamina primordia through a spatially differentially expressed ethylene receptors. For more clarity we rewrote the legend of Figure 6. We also stained in blue the stamina primordia.

9, Fig 3c, how about the ACS11 expression levels in *wip1 ein2* double and single mutant? It would be great if this gene expression is examined to check the model.

ACS11 expression is restricted to flowers on lateral branches. In *wip1 ein2* double and single mutant this expression is not altered. These expressions is shown in Fig3b for the single mutant, and indirectly for the flowers that do not have the WIP and EIN2 functions (CM106-M-LB flowers). For more clarity, we inserted the genotypes of the lines in the panels.

10, In the introduction part, the CmWIP1 may be explained in a bit more detail, e.g., encoding a C2H2 zinc finger-type transcription factor?

For more clarity we inserted the following sentence. 'The G gene codes for CmWIP1, a C2H2 zinc finger transcription factor, which functions as a master regulator in the process of sex determination in cucurbits.'

11, In the discussion part, since the MS is more focused on ethylene signaling components and their regulation on carpel and stamen development, I would suggest that the authors explained the model in the first paragraph. The transposon could be discussed in later part.

Thank you for the suggestion. Indeed, starting with the model ease the reading. We inserted the following paragraph at the start of the discussion:

<<We identified a *PIF/Harbinger* transposon as the origin of the sex transition, disrupting the expression of the Ethylene Insensitive 2 (CmEIN2) gene. Through genetic and expression analyses, we unveiled a dual function of CmEIN2 in sex determination and fruit shape. Upon the expression of CmACS11 in carpel primordia, EIN2 is recruited to suppress the expression of the carpel inhibitor, CmWIP1. Subsequently, the ethylene produced by ACS7 in the carpel primordia is perceived in the stamina primordia through a spatially differentially expressed ethylene receptors¹⁷. EIN2 is then recruited for signaling to inhibit stamina development (Fig. 6).>>

12, In abstract, line 21, 'fruit shape' is better changed to 'fruit shape formation'.

corrected

Reviewer #3 (Remarks to the Author):

This manuscript describes an extensive, comprehensive, and well-performed set of experiments to identify the genetic basis for a spontaneous novel mutation causing conversion to monoecy from androecy in melon. QTL-seq followed by fine-mapping identified a transposon insertion in an ethylene signal transduction gene, EIN2. Ethylene is a known key player in sex expression in cucurbits. The role of EIN2 in the observed phenotypes was verified by identifying and characterizing other EIN2 mutants; and the effect of the mutant on known sex expression genes in melon as well as overall gene expression changes was characterized. The authors then characterized the distribution of this transposon (Harbinger) in the melon genome, identifying both ancient and recent transpositions. Finally, they examined distribution of recent and historical transpositions in cultivated and wild accessions, indicating how transposon activity can be an evolutionary contributor to variation in sex types.

We thank the reviewer for highlighting the quality of our work, and for the corrections that clearly improved the quality of our manuscript.

I only have minor suggestions:

1. Please provide references for the sex phenotype description lines 66-71.

As required by the reviewer, we inserted a review and a recent paper describing the sex phenotype description. The review by Ming et al describes the sex model in melon compared to other species. The Zhang et al paper describes in details melon sex phenotypes, as well as the identification of sex developmental stages.

2. Line 205. "EIN2 promotes the expression of genes associated with cell division and gametophyte development." It seems more likely that this is an indirect effect of promoting carpel development, which in turn leads to the cell division and, especially, gametophyte development rather than a direct effect of EIN2.

We fully agree. EIN2 is an ethylene signaling used as a switch for carpel development and staminal inhibition, so its role is indirect. For more clarity we inserted indirect in the following sense:

<<In summary, our RNA-seq analysis reveals two contrasting indirect roles of CmEIN2 in carpel development, and in staminal inhibition.>>

3. Fig 4 questions:

a. Fig 4 a refers to RefPIF, SANT1, and PIFs identified by TEDseq; however the numbers included within the RefPIF circle do not equal the total provided.

Thank you for pointing this mistake out; we corrected the error. The number of Ref PIF that does not overlap with SANT1 peak, and PIFs identified by TEDseq is 466.

b. Line 220. "Furthermore, 83% of novel insertions locate in promoter regions, compared to 58% of the common ones." Should this reference Fig 4e?

Thanks for reporting this. To properly reference all panels, we have renamed them and checked their in-text citations.

c. The labeling for Fig 4d is confusing. Does 'single peak' refer to the TIR sequence and 'double peak' to the SANT1 sites? If so, it might be helpful to label as those instead. The fact that peaks are single and double is evident from the graph.

DNA transposons of the P Instability Factor (PIF)/Harbinger superfamily are characterized by TIR sequences at their extremities. SANT1 DAP-seq analysis revealed the binding of SANT1 to both TIR

sequences of PIFs, leading to double peaks. When we have double peaks, we are sure 100% that we are detecting a PIF (The two TIR sequences are spaced by a transposase and SANT1 protein).

When we detect only one peak, the situation is more complex, and we have two possibilities: Either we are in a situation of a degenerate PIF that lost one TIR, or a random sequences that has similarities to TIR sequences.

For more clarity we re-annotated all the non-PIF sequences annotated as TE, and analysed further the non-TE bound sequences. We inserted the new panel, as well as the heat map of SANT1 peak density in the region ± 3 kb around all the reference PIFs in Figure S10. We also edited the text in the result section to explain better the DAPseq data.

In the discussion section, we inserted the following section highlighting the efficiency and the limitations of our method: <<Annotating *PIF/Harbinger* elements in plant genomes presents several challenges due to their high diversity in sequences, sizes, and organizations. Beside *in silico* annotation, we used SANT1 ampDAPseq analysis as a tool to identify and annotate *PIF/Harbinger* transposons in melon genomes. In loci displaying ampDAPseq double peaks, we systematically found either a new or an ancient PIF element. However, in loci showing single ampDAPseq peaks, the accuracy of the method is weak, as the binding could be due to degenerate transposon or random sequences that has similarities to TIR sequences. Nevertheless, when we combined transposon annotation with SANT1 ampDAPseq analysis, the approach proved highly efficient in identifying new transpositions as well as ancient and degenerate transpositions. >>

d. Why are there not SANTI peaks for the new insertions shown in Fig 4j?

This normal, the annotated melon genome used for mapping the DAPseq data do not harbors the new insertion, validated by TEDseq analysis.

Minor grammar notes:

Thank you for the corrections. All the spelling mistakes were corrected as suggested

Line 73. 'encodes for' should either be 'encodes' or 'codes for'

Encodes replaced by codes for

Line 91. 'buckets of flowers' should be 'clusters of flowers'

buckets replaced by clusters

Line 92. Write out CRC at first mention

Full name of the gene inserted

Line 124. Delete 'of which'

of which replaced by including

Line 214. 'sensibility' should be 'sensitivity'

sensibility replaced by sensitivity

Line 258. 'constraint' should be 'constrained'

constraint replaced by constrained

Line 295. 'threads' should be 'threats'

threads replaced by threats

REVIEWERS' COMMENTS

Reviewer #1 (Remarks to the Author):

1)

Regarding the transcriptome data of mutant flowers: I admit I originally misunderstood the experiment design used by the authors. In my first review, I recommended to use undifferentiated mutant flower buds instead of differentiated flowers. It turns out that the authors did analyze floral buds, as indicated by the authors, at a stage where "inappropriate sexual organs are arrested".

Therefore, I withdraw my original suggestion to remove this data from the manuscript.

I understand the authors' argument that their design was aimed at investigating the role of EIN2 in staminal inhibition. This does not justify the absence of a WT control, but because the authors mostly interpret GO-term enrichment at clusters, it is acceptable.

Yet, an important criticism I had in the first review still stands. At the end of that result section (lines 261-265), the authors conclude:

a) "In developing carpel, EIN2 promotes the expression of genes associated with cell division and gametophyte development". Could the authors justify their conclusion? In *wip2 ein2* hermaphrodite mutant floral buds, C+ genes are highly expressed, meaning these genes are expressed in the absence of the EIN2 protein, which directly contradicts the authors.

b) That "EIN2 promotes the expression of genes associated with negative regulation and signaling" is directly supported by the data, although the observation would have been more compelling with a female WT flower transcriptome showing expression of the S-cluster.

In summary, I would say that the data is acceptable and support the authors' conclusions, with the exception of a). I suggest to remove a), or provide a justification for this interpretation.

2)

My second major criticism regarded the organization and description of figure 4. The current version is much improved and shows a clear picture of the data.

3)

I thank the authors for thoroughly addressing all my other minor points. Notably, they analyzed non-PIF DAP-seq peaks and provided figures to illustrate their point.

There is an issue with the statistics of figure 5f though: How can one use Mann-Whitney to compare only two data points? This test compares ranks: there is only two data points here (proportion of insertion in promoters: reference versus non-reference)? To the best of my knowledge, the test cannot work in these conditions (or rather, it can only return a p-value of 1).

Two proportions are typically compared with a Z-test or a Fisher exact-test.

Reviewer #2 (Remarks to the Author):

This MS has been improved based on my comments and I have no further comments. However, a minor type error 'ethylene gaz' in the Fig 6 legend should be corrected to 'ethylene gas'.

Reviewer #3 (Remarks to the Author):

The authors have addressed my concerns in the revised manuscript.

Dear reviewers,

We have now revised the last minor comments. We have highlighted in black ink the comments. **Our answers to the points raised are typed in red ink.**

REVIEWERS' COMMENTS

Reviewer #1 (Remarks to the Author):

1)

Regarding the transcriptome data of mutant flowers: I admit I originally misunderstood the experiment design used by the authors. In my first review, I recommended to use undifferentiated mutant flower buds instead of differentiated flowers. It turns out that the authors did analyze floral buds, as indicated by the authors, at a stage where "inappropriate sexual organs are arrested". Therefore, I withdraw my original suggestion to remove this data from the manuscript.

I understand the authors' argument that their design was aimed at investigating the role of EIN2 in stamina inhibition. This does not justify the absence of a WT control, but because the authors mostly interpret GO-term enrichment at clusters, it is acceptable.

Yet, an important criticism I had in the first review still stand. At the end of that result section (lines 261-265), the authors conclude:

a) "In developing carpel, EIN2 promotes the expression of genes associated with cell division and gametophyte development". Could the authors justify their conclusion? In *wip2 ein2* hermaphrodite mutant floral buds, C+ genes are highly expressed, meaning these genes are expressed in the absence of the EIN2 protein, which directly contradicts the authors.

b) That "EIN2 promotes the expression of genes associated with negative regulation and signaling" is directly supported by the data, although the observation would have been more compelling with a female WT flower transcriptome showing expression of the S- cluster.

In summary, I would say that the data is acceptable and support the authors' conclusions, with the exception of a). I suggest to remove a), or provide a justification for this interpretation.

We thank the reviewer for accepting our argument on the transcriptomic design. Concerning the use a WT female flower as control and not a *WIP1* mutant that become female. Technically it is not possible to identify with 100% confidence at stage 6, WT female flowers from male flowers in WT monoecious plants, as they are undistinguishable. Generating male, female and hermaphrodite lines is the best way of sexual flower sampling. This is a standard protocol in the field, and this is why we use female *wip1* flowers as WT control.

Concerning the role of EIN2 in promoting carpel development and inhibiting stamina development, our genetic approach demonstrates that EIN2 is implicated in the inhibition of the carpel inhibitor CmWIP1. By inhibiting *CmWIP* expression, EIN2 promotes the expression of genes associated with cell division and female gametophyte development. In stamina, it is the opposite, EIN2 inhibits stamina development. For more clarity we refers to the figure panels to justify our statements (see paragraph below).

<< In summary, our RNA-seq analysis reveals two contrasting indirect roles of CmEIN2 in carpel development, and in stamina inhibition. In developing carpel, through inhibition of the carpel repressor, *CmWIP1* (Fig. 3b), EIN2 promotes the expression of genes associated with cell division and female gametophyte development, including the carpel identity gene, *CRC* (Fig. 3g, Supplementary Fig. S6e). In inhibited stamina, EIN2 promotes the expression of genes associated with negative regulation and signaling, and the repression of the stamina promoting gene, *PI* (Fig. 3g, Supplementary Fig. S6c). >>

2)

My second major criticism regarded the organization and description of figure 4. The current version is much improved and shows a clear picture of the data.

Thank you for the suggestions that clearly improved Figure 4.

3)

I thank the authors for thoroughly addressing all my other minor points. Notably, they analyzed non-PIF DAP-seq peaks and provided figures to illustrate their point.

There is an issue with the statistics of figure 5f though: How can one use Mann-Whitney to compare only two data points? This test compares ranks: there is only two data points here (proportion of insertion in promoters: reference versus non-reference)? To the best of my knowledge, the test cannot work in these conditions (or rather, it can only return a p-value of 1).

Two proportions are typically compared with a Z-test or a Fisher exact-test.

Thank you for pointing this. We performed a Z-test for these data.

Reviewer #2 (Remarks to the Author):

This MS has been improved based on my comments and I have no further comments. However, a minor type error 'ethylene gaz' in the Fig 6 legend should be corrected to 'ethylene gas'.

Thank you for pointing this, we corrected the spelling mistake.

Reviewer #3 (Remarks to the Author):

The authors have addressed my concerns in the revised manuscript. Thank you for the review.